# Epstein-barr virus latent membrane protein 1 targets cIAP1, cIAP2 and TRAF2 for proteasomal degradation to activate the non-canonical NF-κB pathway

Yizhe Sun[1,2,3☉]*, Shunji Li[1,2,3☉], Bidisha Mitra[1,2,3], Ling Zhong[1,2,3], Aretina Zhang[1,2,3], Benjamin E. Gewurz [1,2,3]*

**1** Division of Infectious Diseases, Department of Medicine, Brigham and Women's Hospital, Boston, Massachusetts, United States of America, **2** Center for Integrated Solutions for Infectious Diseases, Broad Institute, Cambridge, Massachusetts, United States of America, **3** Department of Microbiology, Harvard Medical School, Boston, Massachusetts, United States of America

☉ These authors contributed equally to this work.
* bgewurz@bwh.harvard.edu (BEG); ysun34@bwh.harvard.edu (YS)

## Abstract

The Epstein-Barr virus (EBV) oncoprotein Latent Membrane Protein 1 (LMP1) is expressed in multiple malignancies and is critical for B-cell immortalization. LMP1 constitutively activates NF-κB signaling pathways, which are essential for EBV-mediated B cell transformation and for transformed B cell survival. Reverse genetic analysis revealed two LMP1 regions critical for primary human B cell immortalization, termed transformation effector site (TES) 1 and 2, which activate multiple host growth and survival pathways, in particular NF-κB. Of these, only TES1 signaling is required for B-cell transformation within the first several weeks of infection. TES1 signaling is also critical for EBV-transformed lymphoblastoid B-cell survival. However, precisely how TES1 initiates NF-κB signaling has remained incompletely understood. Here, we provide multiple lines of evidence that TES1 associates with cellular inhibitor of apoptosis protein 1 and 2 (cIAP1/2) in a tumor necrosis factor associated factor 3 (TRAF3) dependent manner. TES1 signaling drives cIAP1 autoubiquitination and targets TRAF2, cIAP1 and 2 for proteasomal degradation in a TRAF3 dependent manner. Overexpression of either cIAP1 or 2 impaired LMP1 TES1-mediated non-canonical NF-κB activation. Collectively, these studies suggest that LMP1 TES1 initiates non-canonical NF-κB signaling distinctly from CD40 and other host immunoreceptors, thereby highlighting a therapeutic target.

## Author summary

EBV is a common human virus that can drive the growth and survival of infected B cells, contributing to multiple kinds of cancers. A key viral oncoprotein called

**Data availability statement:** The authors confirm that all data underlying the findings are fully available without restriction. All relevant data are within the manuscript and its Supporting Information files.

**Funding:** This work was supported by the National Institutes of Health (NIH) R01CA228700 and R01DE033907 to BEG, American Cancer Society Post-doctoral Fellowship (PF-23-1144614-01-IBCD to YS; PF-24-1250090-01-IBCD to BM), and Harvard University Center for AIDS Research (CFAR) NIH funded program (P30 AI060354 to SL). The Harvard University Center for AIDS Research (CFAR) NIH funded program is supported by the following NIH Co-Funding and Participating Institutes and Centers: NIAID, NCI, NICHD, NHLBI, NIDA, NIMH, NIA, NIDDK, NIMHD, NIDCR, NINR, OAR, and FIC. The authors also acknowledge generous support from George and Sandra K Schussel. The funders had no role in study design, data collection and analysis, decision to publish, or preparation of the manuscript.

**Competing interests:** The authors have declared that no competing interests exist.

LMP1 is central to this process, because it activates the non-canonical and canonical NF-κB pathways, which upregulate genes that drive B cell proliferation. Two LMP1 C-terminal cytoplasmic tail regions, TES1 and TES2, are genetically defined to be critical for EBV-mediated B-cell immortalization. TES1 is essential during the earliest stages of infection and for keeping EBV-transformed B cells alive, and is critical for LMP1-mediated non-canonical NF-κB activation. However, how TES1 triggers non-canonical NF-κB signaling has remained incompletely understood. In this study, we found that TES1 interacts with two host proteins, cIAP1 and cIAP2, through another signaling molecule, TRAF3. TES1 activity causes cIAP1 to modify itself and leads to the degradation of TRAF2, cIAP1, and cIAP2 to activate the non-canonical NF-κB pathway. We also found that increasing the amount of cIAP1 in cells blocks TES1's ability to activate this pathway. These findings reveal that EBV uses a mechanism distinct from B cell receptors to turn on the non-canonical NF-κB pathway, highlighting a unique viral strategy and a potential therapeutic target for EBV-associated cancers.

## Introduction

The gamma-herpesvirus Epstein-Barr virus (EBV) establishes lifelong infection of nearly 95% of adults worldwide. EBV causes ~200,000 cancer cases per year, including multiple types of lymphomas, gastric and nasopharyngeal carcinoma [1–4]. These include Burkitt lymphoma, Hodgkin lymphoma, primary central nervous system (CNS) lymphoma, post-transplant lymphoproliferative disease (PTLD), T and NK cell lymphoma [5]. EBV is also a major trigger of autoimmune diseases, in particular multiple sclerosis and systemic lupus erythematosus [6–9].

EBV subverts key host pathways to drive proliferation and differentiation of newly-infected B cells into memory cells, the reservoir for lifelong infection. To do so, EBV expresses a series of latency oncogenes, comprised of six Epstein-Barr nuclear antigens (EBNA) and two latent membrane proteins (LMP), together with non-coding RNAs (ncRNA). Latency programs utilize distinct combinations of viral latency genes [10–13]. The fully transforming latency III program, comprised of all six EBNA, LMP1, LMP2A, LMP2B and ncRNAs immortalizes infected cells into lymphoblastoid B cell lines (LCLs). Latency III cells are observed in PTLD and primary CNS lymphoma. Another latency program, seen in EBV-infected memory B cells and Burkitt lymphoma cells, is latency I, in which EBNA1 is the only viral protein expressed and LMP1 is not expressed.

LMP1 expression is sufficient to drive rodent fibroblast transformation and poly-clonal B-cell proliferation [14–16]. LMP1 is comprised of a 24-residue cytoplasmic N-terminal tail, six transmembrane domains and a 200 residue C-terminal cytoplasmic tail [11,13,17]. LMP1 transmembrane domains drive lipid raft association and continuous, ligand independent signaling from C-terminal tail regions [18–21]. Of these, reverse genetic studies defined that signaling from transformation effector sites (TES) 1 and 2, also referred to as C-terminal activating regions (CTAR) 1 and 2, are necessary for EBV-mediated primary B-cell transformation [18,22].

TES1 spans residues 186–231 and uses a PXQXT motif to engage tumor necrosis factor receptor associated factors (TRAFs) to drive downstream non-canonical NF-κB, MAP kinase and PI3K pathways [23–27]. TRAF1 enables TES1 to also potently activate canonical NF-κB [26]. TES2 residues 351–386 recruit TRAF6 [28] and independently triggers canonical NF-κB, MAPK, IRF7, and P62 pathways [29,30]. The LMP1 C-terminal tail CTAR3 region also activates JAK/STAT and SUMOylation pathways [31]. Of these, only TES1 signaling is necessary for EBV B cell transformation [18,22–27,32,33].

LMP1 mimics aspects of signaling by the B-cell co-receptor CD40, whose binding by CD40 ligand stimulates NF-κB, MAP kinase activity. CD40 signaling is required for B-cell development, germinal center formation, class switch recombination and somatic hypermutation [34,35]. Transgenic mice that express the LMP1 C-terminal tail in place of that of CD40 have normal B-cell development, activation and immune responses, though do exhibit cytokine-independent class-switch recombination [36]. Similarly, mice with transgenic B-cell LMP1 or chimeric LMP1/CD40 cytoplasmic tail have largely overlapping phenotypes [37]. However, since LMP1 signals constitutively and in a ligand-independent manner, its expression is transforming. Transgenic LMP1 expression transforms rodent fibroblasts [14]. In the absence of T and NK surveillance, transgenic LMP1 expression drives fatal B-cell proliferation, particularly when co-expressed with LMP2A, which mimics aspects of B-cell immunoglobulin signaling [12,38,39]. LMP1 and EBNA2 co-expression are sufficient to transform human peripheral blood B cells into immortalized lymphoblasts [40], further suggesting LMP1 as a major EBV therapeutic target.

The non-canonical NF-κB pathway promotes B cell survival, maturation and homeostasis in peripheral lymphoid organs, and has key roles in germinal center formation [41,42]. Impairment of non-canonical NF-κB signaling results in immunodeficiency, whereas its hyperactivation instead contributes to autoimmune, inflammatory and neoplastic diseases [41]. Non-canonical NF-κB is suppressed at baseline by constitutive targeting of the NF-κB activating kinase (NIK) for ubiquitin-mediated proteasomal degradation. A complex containing TRAFs 2, 3, cellular inhibitor of apoptosis (cIAP) 1 and 2 binds to NIK, typically using TRAF3 as an adaptor protein [41,42]. The E3 ubiquitin ligase activity of cIAP 1 and 2 targets NIK for proteasomal degradation to suppress non-canonical NF-κB signaling in the absence of stimuli.

Through multiple distinct mechanisms, non-canonical pathways disrupt cIAP-mediated NIK turnover to activate downstream signaling. For instance, to initiate signaling, CD40 and B-cell activating factor (BAFF) receptors trigger cIAPs to ubiquitinate and target TRAF3 for proteasomal degradation, thereby preventing NIK ubiquitination by the TRAF2/cIAP1/cIAP2 complex. To do so, they increase K63-linked polyubiquitin chain attachment to TRAF2 [43,44]. Increased NIK abundance results in its autophosphorylation, which stimulates its kinase activity towards the kinase IKKα. Activated IKKα phosphorylates p100, which triggers its partial proteasomal processing into the active p52 NF-κB transcription factor subunit. p52 traffics into the nucleus as homodimers or heterodimers, including with RelB, to activate non-canonical pathway targets [41]. Comparatively less is known about how TES1 initiates non-canonical NF-κB signaling.

The LMP1 PXQXT motif acts as a docking site for TRAFs 1, 2, 3 and 5 [32,45–47]. A substantial amount of the LCL TRAF1 and 3 pools are associated with LMP1, whereas a comparatively low amount of TRAF2 is LMP1 associated [24,48–50]. TRAF3 binds more tightly to LMP1 than other TRAFs and can compete with TRAF1 and TRAF2 for LMP1 association [24,48,49]. It has been suggested that TES1 sequesters TRAF3 to initiate non-canonical NF-κB signaling [51]. LMP1 signaling was intact in TRAF2-/- B cells [46,52], though TRAF2 knockout itself activates non-canonical NF-κB, complicating these analyses. TES1 signaling activates non-canonical NF-κB through NIK and IKKα [25,53–57].

A range of LMP1 target genes have been defined [58,59]. Key TES1 targets include the anti-apoptotic factor cFLIP, which is necessary for LCL survival [27,60], underscoring the importance of TES1 signaling. Yet, the mechanism by which LMP1 initiates non-canonical NF-κB signaling has remained incompletely defined. Interestingly, despite its central role in control of other non-canonical pathways, cIAP activity has not been investigated in the context of LMP1 signaling. Here, we report that LMP1 associates with both cIAP1 and 2, in a TRAF-dependent manner. We present evidence that LMP1 recruits and targets cIAP1, 2 and TRAF2 for proteasomal degradation in a TRAF3-dependent manner to initiate non-canonical NF-κB signaling, suggesting a previously unappreciated manner by which LMP1 TES1 drives this key oncogenic pathway.

                                                                                            

## Results

### LMP1 TES1 signaling downmodulates TRAF2, cIAP1 and cIAP2 abundances

To gain insights into how LMP1 initiates non-canonical NF-κB signaling, we used EBV-negative Daudi and Akata Burkitt B-cell lines with doxycycline inducible expression of wildtype (WT) vs well-characterized LMP1 TES1 vs TES2 mutant alleles, in which point mutations abrogate signaling from TES1 and/or TES2 [27,61,62]. The TES1 mutant (TES1m) harbored two alanine point mutations in the TRAF binding motif (204PQQAT208→AQAAT) that impede TRAF recruitment, whereas the TES2 mutant (TES2m) 384YYD386→ID substitutions block TES2 signaling. We also utilized an LMP1 TES1 and TES2 double mutant (DM) with both of these mutations (Fig 1A). We validated that each had similar levels of LMP1 expression at 24 hours post-induction by 250 ng/ml doxycycline, that LMP1 with signaling from either or both TES induced TRAF1 albeit to varying degrees as expected [27], and that non-canonical NF-κB activation was highly impaired by TES1m or DM LMP1, as expected (Fig 1B).

While cIAP ubiquitin ligases have major roles in control of most non-canonical NF-κB pathways [41], they have not yet been characterized downstream of LMP1. We were therefore intrigued to find that cIAP1 and 2 were highly depleted in both Daudi and Akata B cells 24 hours after induction of expression of WT or TES2m LMP1, in which TES1 signaling was active. By contrast, we did not observe significantly changed cIAP1 or 2 expression following induction of similar levels of TES2m or DM LMP1, in which TES1 signaling was inactive (Figs 1B–1C and S1A–S1B). Notably, levels of inhibitor of apoptosis family member XIAP remained unchanged upon expression of any of the LMP1 constructs (Figs 1B and S1A), suggesting a potentially specific TES1 effect at the level of cIAP1 and 2. We also noticed that induction of WT or TES2m LMP1 resulted in diminished levels of TRAF2, but not TRAF3 (Figs 1B–1C and S1A–S1B). Furthermore, we noted that a lower molecular weight band immunoreactive with anti-TRAF2 antibody appeared following induction of WT or TES2m expression (Figs 1B and S1A), suggesting that TES1 signaling may induce TRAF2 cleavage. We note that CD40 signaling does not induce cIAP1/2 degradation (S1C Fig) [65,66].

As a complementary approach, we assessed cIAP1 and 2 levels in primary human B-cells infected by recombinant EBV with wildtype, TES1m, TES2m or DM LMP1. Of note, LMP1 is not required for the first 8 days of newly infected primary B-cell outgrowth [67]. We observed significantly lower cIAP1 and cIAP2 levels in cells infected by EBV encoding WT or TES2m LMP1, in which TES1-mediated non-canonical NF-κB signaling was active, as evidenced by robust p100:p52 processing (Fig 1D–1E). Taken together, these results raise the possibility that TES1 signaling may target cIAP1 and cIAP2 for degradation.

### Latency III destabilizes TRAF2, cIAP1 and cIAP2 in B-cells

Constitutive LMP1/NF-κB signaling in latency III cells drives synthesis of both cIAP1 and cIAP2, which are well-characterized NF-kB target genes [68]. In contrast to the loss of cIAP1/2 and TRAF2 seen upon induction of conditional LMP1 expression in Burkitt cells, expression of each is detectable in latency III Burkitt cells and in LCLs. We therefore hypothesized that their steady state levels in latency III cells represent the balance of their synthesis and turnover. To test this, we performed cycloheximide (CHX) chase analysis, using isogenic MUTU Burkitt cells with latency I vs III programs, termed MUTU I vs III [69]. LMP1 displayed a half-life of 4 hours, consistent with previous research [70]. Interestingly, cIAP1 and cIAP2 exhibited shorter half-lives in MUTU III than in MUTU I, with $T_{1/2} < 1$ hour in MUTU III but $> 4$ hours in MUTU I (Fig 2A–2B). Similarly, TRAF2 levels decreased by ~50% over the first four hours of cycloheximide chase in MUTU III, but were little changed in MUTU I at this time point (Fig 2A–2B). Intriguingly, we observed two TRAF2 species in immunoblots of MUTU III whole cell lysates (WCL), which migrated at a similar molecular weight and at a somewhat lower molecular weight than in blots of MUTU I WCL (Fig 2A). These data are consistent with a model in which LMP1 targets cIAP1/2 for degradation, but also induces cIAP1/2 expression through NF-κB target gene regulation, potentially in a manner dependent on TRAF2 proteolysis.

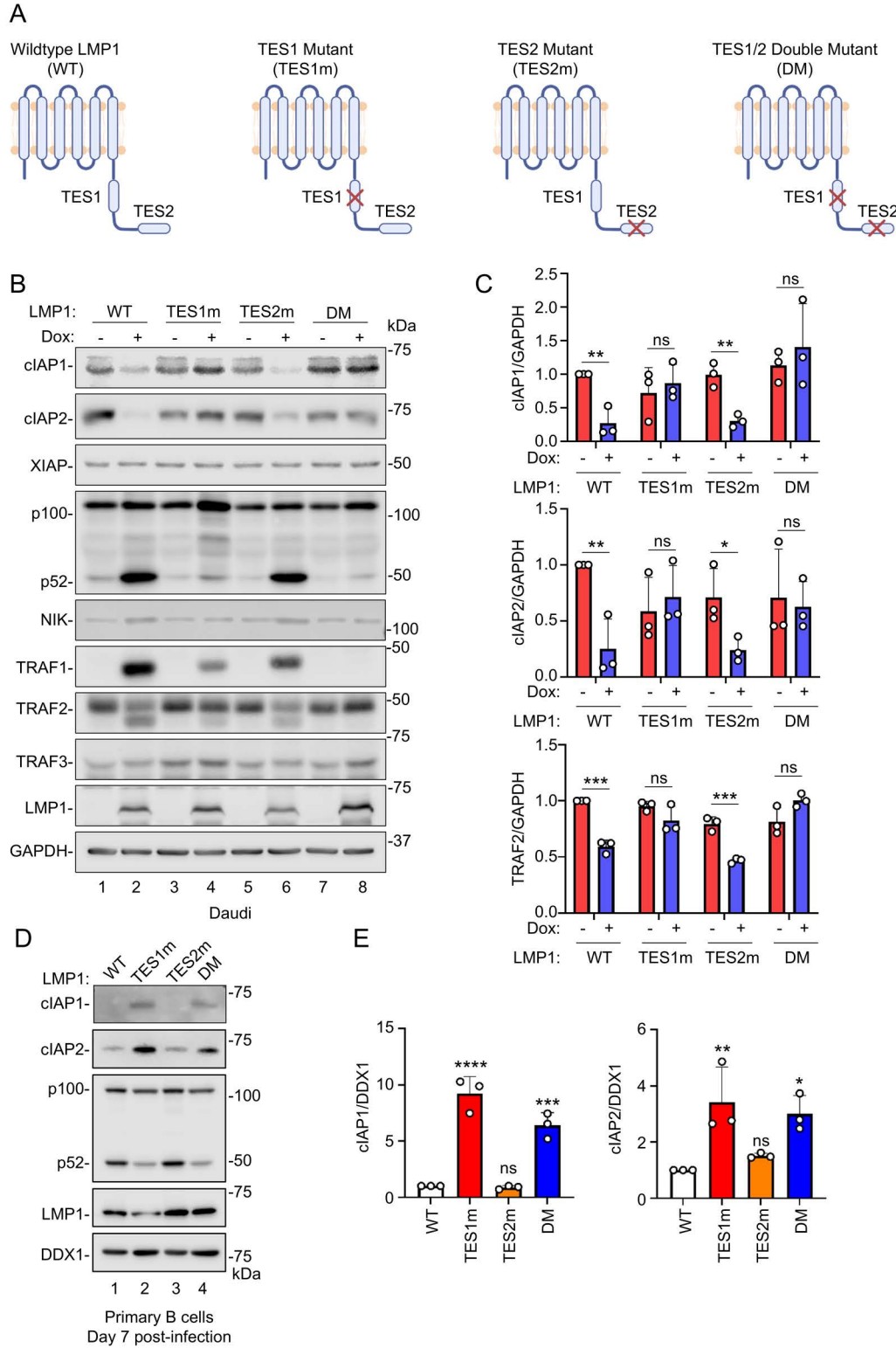

**Fig 1. LMP1 TES1 signaling downregulates cIAP1, cIAP2 and TRAF2 expression. (A)** Schematic diagram of wildtype (WT), TES1 point mutant (TES1m), TES2 point mutant (TES2m), or TES1/2 double mutant (DM) LMP1. The TES1 point mutation selectively abrogates TES1 signaling while maintaining TES2 signaling; the TES2 point mutation selectively abrogates TES2 signaling while preserving TES1 signaling; DM TES1 and TES2 point mutations abrogate signaling from both TES1 and TES2. Created in BioRender. Sun, Y. (2026) https://BioRender.com/xa5k6rv. **(B)** Analysis of LMP1

effects on cIAP1, cIAP2, XIAP, NIK and TRAF expression in Daudi Burkitt cells. Immunoblot analysis of whole cell lysates (WCL) from Cas9+Daudi cells induced for WT, TES1m, TES2m or DM LMP1 expression by addition of 250ng/mL doxycycline (Dox) for 24 hours. **(C)** Relative fold changes+standard deviation (SD) of GAPDH load control normalized cIAP1, cIAP2 and TRAF2 levels, based on densitometry from three replicates as in **(B)**. Values in vehicle control treated WT LMP1 expressing cells were set to 1. **(D)** Analysis of LMP1 effects on cIAP1 and cIAP2 expression in newly infected primary B-cells. Immunoblot analysis of WCL from human peripheral blood B cells infected with EBV expressing WT, TES1m, TES2m or DM LMP1 at day 7 post-infection. DDX1 was used as a load control, since its protein levels do not change significantly following primary B cell infection by EBV [63,64]. **(E)** Relative fold changes+SD of DDX1 load control normalized cIAP1 and cIAP2 levels, based on densitometry from three replicates as in **(D)**. Values in vehicle control treated WT LMP1 expressing cells were set to 1. Statistical significance was assessed by two-tailed unpaired Student's t-test (C) or one-way ANOVA followed by Tukey's multiple comparisons test **(E)**. ns, not significant, *$p < 0.05$, **$p < 0.01$, ***$p < 0.001$, ****$p < 0.0001$. Blots (B and D) are representative of n=3 experiments.

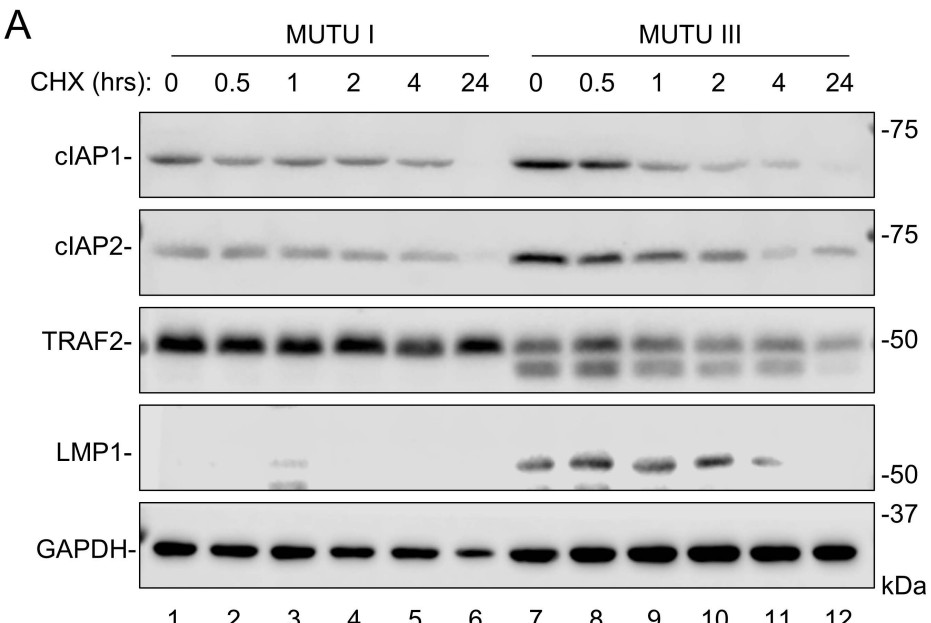

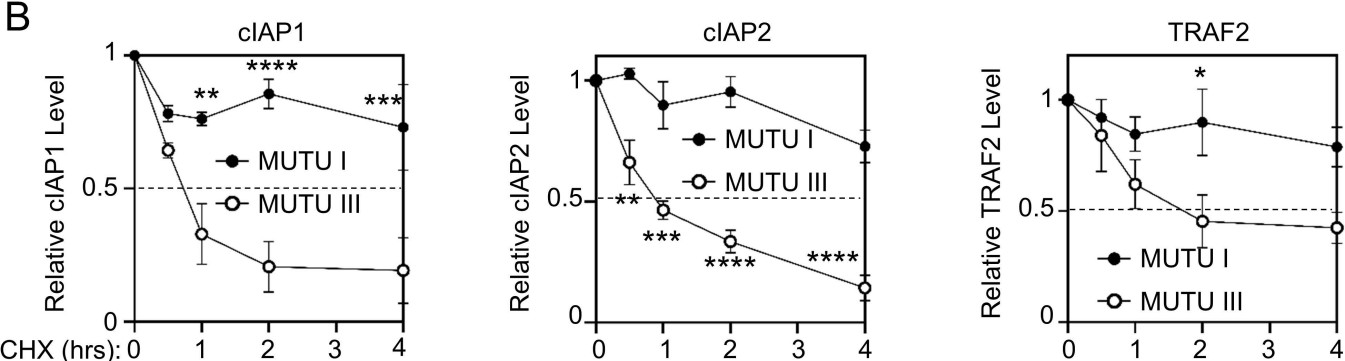

**Fig 2. EBV latency III destabilizes cIAP1, cIAP2 and TRAF2. (A)** Analysis of cIAP1, cIAP2, and TRAF2 half-lives in isogenic MUTU Burkitt B-cells that differ only by EBV latency I vs III programs. Shown are representative immunoblot analyses of WCL from MUTU I vs MUTU III cells treated with 50 µg/mL cycloheximide (CHX) for the indicated number of hours. Blots are representative of n=3 experiments. **(B)** Relative fold changes+SD of GAPDH load control normalized cIAP1, cIAP2 and TRAF2 levels, based on densitometry from three replicates as in **(A)**. Values in MUTU I and III cells at 0 hours of CHX chase were set to 1. Statistical significance was assessed by two-tailed unpaired Student's t-test **(B)**. ns, not significant, *$p < 0.05$, **$p < 0.01$, ***$p < 0.001$, ****$p < 0.0001$.

## TRAF3 is required for LMP1-mediated cIAP1/2 degradation

Given that TES1 recruits multiple TRAFs [24], and that particular TRAFs can recruit cIAPs to immunoreceptor signaling regions [71,72], we next asked whether specific TRAFs were necessary for TES1 targeting of cIAP1/2. We therefore used CRISPR-Cas9 editing to selectively deplete either TRAF1, 2, 3 or 5, each of which associate with the TES1 PxQxT motif [24,48,49,73–76]. Following single guide RNA (sgRNA) expression in Cas9 + Daudi Burkitt B-cells, immunoblot was used to confirm successful on-target TRAF depletion. We also confirmed that TRAF3 depletion was sufficient to trigger non-canonical NF-κB activity, as judged by p100:p52 processing (Fig 3A). Intriguingly, depletion of TRAF3, but not of TRAFs 1, 2 or 5, significantly stabilized cIAP1 and cIAP2 upon conditional WT LMP1 expression (Figs 3A–3B and S2). As expected, TES1m LMP1 expression failed to deplete cIAP1 or 2 in cells depleted of TRAF1,2,3 or 5 (Figs 3A–3B and S2). Since TRAF2 and 5 have partially redundant function [77–80], it was however notable that conditional WT, but not TES1m LMP1 expression, also partially depleted TRAF5 (S2C Fig).

To further assess TRAF3 roles in cIAP1/2 turnover in the latency III context, we next performed cycloheximide chase analysis in the well-characterized LCL GM12878. CRISPR TRAF3 depletion again stabilized cIAP1 (Fig 3C), even though TRAF3 itself exhibited remarkable stability over the 24-hour chase in control cells. TRAF3 depletion also prolonged TRAF2 half-life (Fig 3C–3D), suggesting that TES1 signaling may target a complex containing TRAF2, potentially also TRAF5, cIAP1 and cIAP2 for degradation in a TRAF3 dependent manner.

## LMP1 association with cIAP1/2 is dependent on TRAF3, but not TRAFs1, 2 or 5

We next examined whether LMP1 associated with cIAP1/2 in B-cells in a TES1-dependent manner. C-terminally HA-tagged LMP1 co-immunoprecipitated TRAFs 2 and 3, as expected, but also co-immunoprecipitated cIAP1 and cIAP2 (Fig 4A). Importantly, TES1m and DM LMP1 failed to co-immunoprecipitate TRAFs 2/3 or cIAP1/2, suggesting that their association with LMP1 was dependent on the TES1 PxQxT motif (Fig 4A). Similarly, confocal immunofluorescence analysis demonstrated LMP1 and cIAP1 colocalization, which was again dependent on an intact TES1 PxQxT sequence (S3 Fig).

As TRAFs recruit cIAP1/2 to receptors [42,43,71,81], we next examined whether specific TRAFs were necessary for cIAP1/2 association with LMP1. CRISPR TRAF3 depletion strongly diminished cIAP1 association and to a somewhat lesser extent cIAP2 association with HA-LMP1 in Daudi cell extracts (Fig 4B). Importantly, TRAF3 depletion did not impair recruitment of TRAF2, which associated with LMP1 in a TES1 PxQxT motif-dependent fashion (Fig 4B). By contrast, CRISPR depletion of either TRAFs 1, 2 or 5 did not substantially diminish cIAP1 or cIAP2 co-immunoprecipitation with HA-LMP1 (Fig 4C). We extended this result into the latency III setting, using Cas9 + LCLs that express an N-terminally FLAG-tagged LMP1 allele knocked into the EBV genome [48]. FLAG-LMP1 co-immunoprecipitated cIAP1, 2 and TRAF2 in a manner dependent on TRAF3, but not TRAFs1, 2 or 5 (Fig 4D). By contrast, TRAF2 associated with LMP1 in control LCLs or in LCLs depleted for TRAFs 1, 3 or 5, suggesting specificity of the cIAP1/2 result. Altogether, these data suggest that LMP1 TES1 associates with TRAF3- containing TRAF heterotrimers, which likely also include TRAF2 and which in turn recruit cIAP1 and cIAP2.

## LMP1 TES1 triggers cIAP1 polyubiquitination and cIAP1/2 proteasomal degradation

To gain insights into how TES1 signaling results in TRAF2 and cIAP1/2 turnover, we performed immunoblot analysis on whole cell lysates from Daudi cells treated with vehicle, the proteasome inhibitor MG132 or with the lysosome acidification inhibitor NH$_4$Cl followed by induction of WT or TES1m LMP1. TES1-dependent depletion of cIAP1 and cIAP2 was evident in vehicle control and NH$_4$Cl-treated cells, but to a considerably lesser degree in MG132-treated cells (Figs 5A and S4A). MG132 mildly stabilized TRAF2, with the majority of TRAF2 species shifting to a lower molecular weight in control and MG132-treated Daudi cells induced for WT LMP1, but not for TES1m LMP1 expression (S4B Fig). These results suggest that partial proteasomal digestion does not result in the modified TRAF2 species. Antibody mapping assays indicated that TRAF2 N-terminal cleavage accounts for the lower-molecular weight TRAF2 band (S4C Fig), which had similar

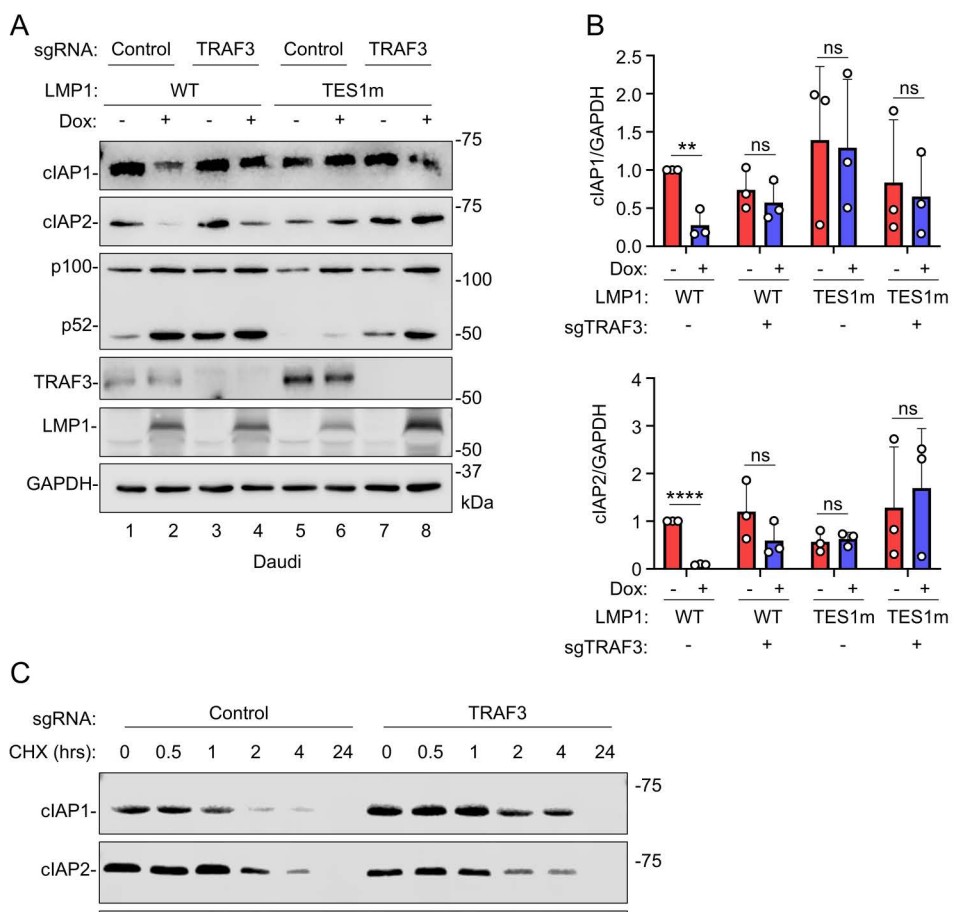

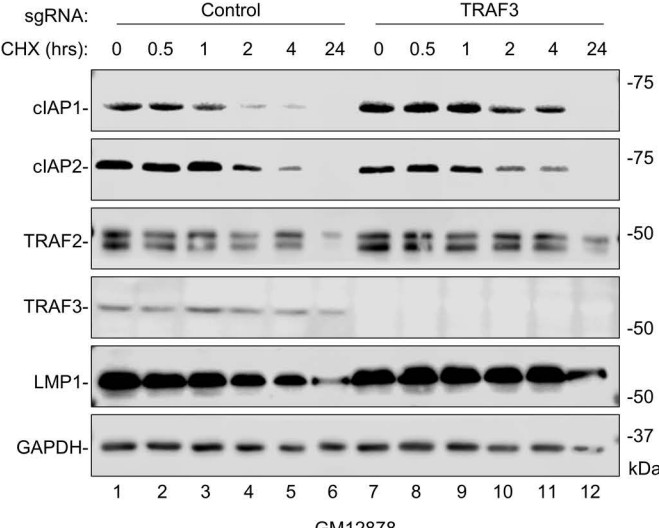

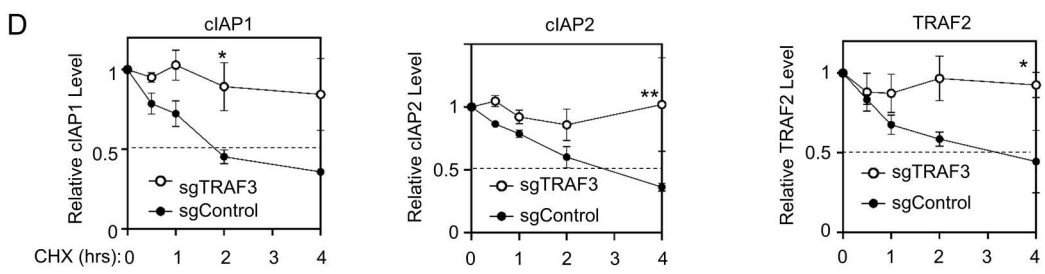

**Fig 3. TRAF3 supports LMP1-induced cIAP1/2 depletion. (A)** Analysis of TRAF3 roles in LMP1 TES1-mediated cIAP1 and cIAP2 depletion. Immunoblot analysis of WCL from Cas9+Daudi cells that expressed control vs TRAF3 targeting single guide RNA (sgRNA) and that were induced for LMP1 expression by Dox (250 ng/mL) for 24 hours. **(B)** Relative fold changes+SD of GAPDH load control normalized cIAP1, cIAP2, based on densitometry from three immunoblot replicates as shown in **(A)**. Values in Daudi cells with control sgRNA and uninduced for WT LMP1 expression were set to 1. **(C)**

Analysis of cIAP1, cIAP2, and TRAF2 half-lives in control vs TRAF3 knockout (KO) GM12878 LCLs. Immunoblot analysis of WCL from Cas9+GM12878 LCLs that expressed control versus TRAF3 targeting sgRNA and that were treated with 50μg/mL CHX for the indicated hours (hrs). **(D)** Relative fold changes+SD of GAPDH load control normalized cIAP1, cIAP2 and TRAF2 levels, based on densitometry from three replicates as in **(C)**. Values in GM12878 cells with control sgRNA expression at 0 hours of CHX chase were set to 1. Statistical significance was assessed by two-tailed unpaired Student's t-test **(B,D)**. ns, not significant, *p<0.05, **p<0.01, ****p<0.0001. Blots are representative of n=3 experiments.

electrophoretic mobility to an N-terminal TRAF2 deletion mutant lacking the first 60 residues (S4D Fig). Taken together, these results suggest that LMP1 TES1 signaling triggers TRAF2 cleavage within the N-terminal RING domain.

cIAP1 and cIAP2 are E3 ubiquitin ligases, and their activity can play non-redundant versus redundant roles on downstream signaling pathways, in particular non-canonical NF-κB activation [82]. cIAP1 RING domain sequestration serves as an important mechanism of its E3 autoinhibition (Fig 5B), which can be relieved by small molecule second mitochondria-derived activator of caspases (SMAC) mimetics [83–85]. We therefore next characterized how TES1 signaling alters cIAP1 vs cIAP2 poly-ubiquitination. Since the cIAP1 CARD-RING domains is able to downmodulate protein levels of both cIAP1 and cIAP2 [86], we first examined whether LMP1 TES1 signaling triggers polyubiquitin chain attachment to cIAP1 molecules. To do so, we immunoprecipitated endogenous cIAP1 from Daudi cells mock-induced or induced for WT vs TES1m LMP1 for 24 hours, using whole cell lysates that were boiled for 5 minutes to disrupt protein-protein complexes. Intriguingly, abundant poly-ubiquitin chain modification was evident on cIAP1 immunoprecipitated from WT LMP1 expressing cells, but not from the mock induced or from TES1m expressing cells (Fig 5C).

To build upon this result, we next co-expressed either HA-tagged wildtype or RING mutant cIAP1, in which an H588A point mutation abrogates cIAP1 activity [87] (Fig 5B). 293T were then co-transfected with empty vector or expression constructs for WT, TES1m, TES2m or DM LMP1, together with HA-tagged wildtype vs RING mutant cIAP1 for 24 hours. Cells were then treated with MG132 for another 24 hours to inhibit proteasomal turnover. Anti-HA immunoprecipitation was performed on whole cell lysates that were boiled for 5 minutes, followed by immunoblot for poly-ubiquitin chains, which revealed that WT and TES2m LMP1, but not TES1m or DM LMP1, triggered abundant HA-cIAP1 poly-ubiquitin chain decoration. Interestingly, LMP1 failed to induce RING mutant HA-cIAP1 polyubiquitination (Fig 5D). Similar results were observed in Daudi cells (S4E Fig).

We then cross-compared LMP1 TES1 and TES2 signaling effects on cIAP1 vs cIAP2 polyubiquitination. 293T were co-transfected with constructs encoding WT vs point mutant LMP1, together with HA-cIAP1 or cIAP2 expression vectors. MG132 was again added 24 hours post-transfection for a further 24 hours. HA-cIAP1 or cIAP2 were then immunoprecipitated from boiled whole cell lysates and subjected to polyubiquitin immunoblot. WT and TES2m LMP1 stimulated poly-ubiquitin chain attachment to HA-cIAP1, but to a considerably lesser degree to HA-cIAP2, as judged by quantitation of the poly-ubiquitin to HA immunoblot signal (Fig 5E). Therefore, our data suggests that TES1 signaling triggers cIAP1 polyubiquitination, potentially via autoubiquitination. Co-transfection of cIAP1 and cIAP2 cDNAs triggered robust cIAP2 polyubiquitination in a TES1-dependent manner (Fig 5F), indicating that cIAP1 likely supports LMP1-mediated cIAP2 polyubiquitination, potentially as the ubiquitin ligase. While 293T also expresses endogenous cIAP1, we hypothesize that cIAP1 overexpression was required to stoichiometrically match transfected cell cIAP2 levels. By contrast, cIAP2 knockout did not strongly affect TRAF2 cleavage, cIAP1 depletion or p100:p52 processing in Daudi cells (S4F Fig). Taken together, these results support a model in which LMP1 TES1 drives cIAP1 autoubiquitination to drive its depletion and potentially also that of TRAF2 and cIAP2.

## TRAF3 is necessary for TES1-driven cIAP1 polyubiquitination

We hypothesized that TRAF3 is critical for cIAP1 polyubiquitination downstream of LMP1, given its role in cIAP1/2 recruitment to TES1. To test this, we first transfected 293T control versus TRAF2, 3 or 5 KO 293 cells with expression vectors encoding HA-cIAP1, alone or together with V5-tagged LMP1. We then immunoprecipitated HA-cIAP1 from boiled lysates,

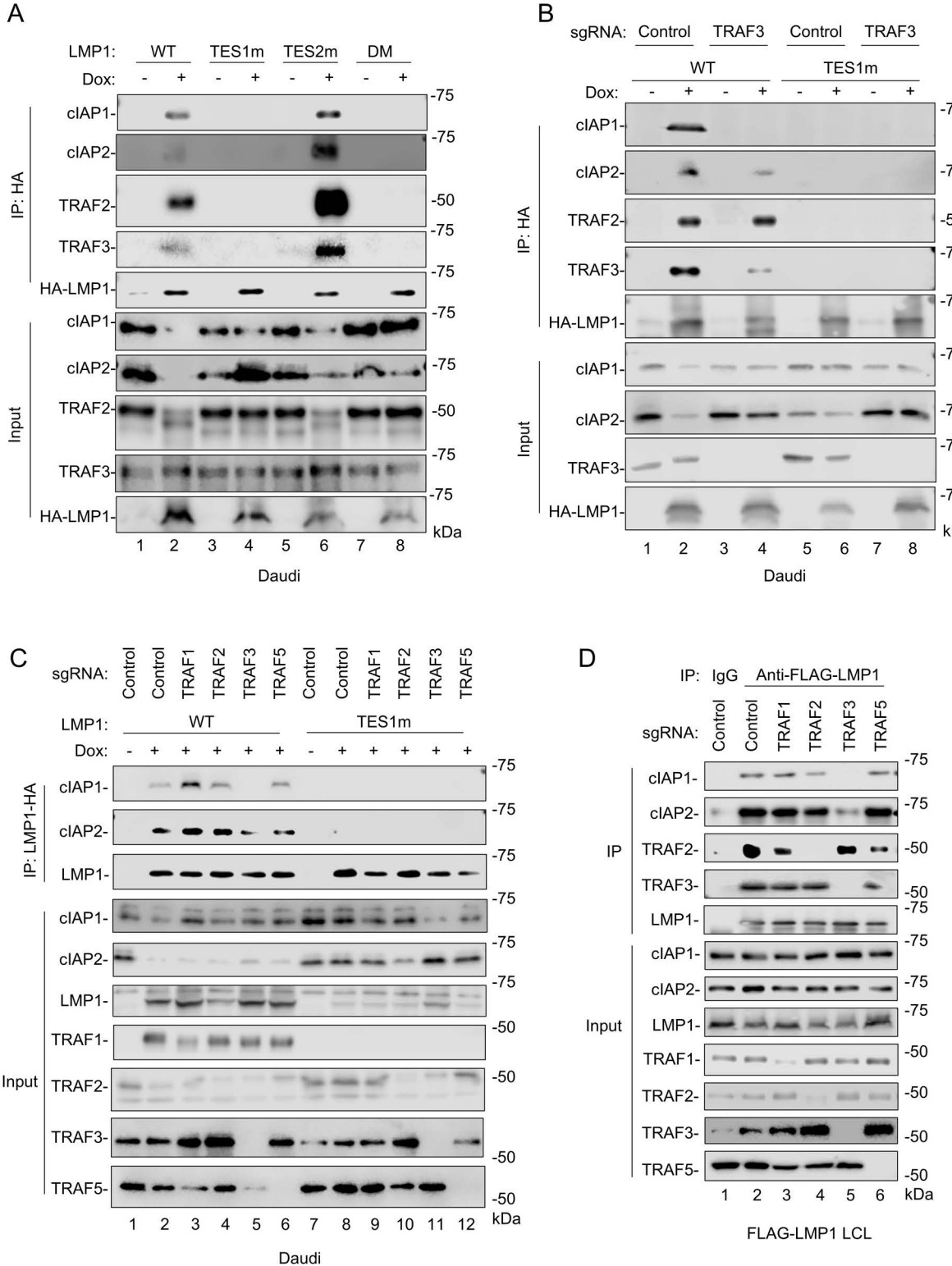

**Fig 4. TRAF3 supports LMP1 association with cIAP1 and 2 in a TES1-dependent manner. (A)** Analysis of LMP1 TES1 versus TES2 association with cIAP1 and cIAP2. Immunoblot analyses of 1% input vs anti-HA-LMP1 complexes immunopurified from Daudi cells mock induced or induced for HA-tagged WT, TES1m, TES2m or DM LMP1 for 24 hours by 250 ng/ml Dox. **(B)** Analysis of TRAF3 role in LMP1 association with cIAP1/2. Immunoblot

analyses of 1% input vs anti-HA-LMP1 complexes immunopurified from Cas9+Daudi cells that expressed control versus TRAF3 targeting sgRNA and induced for WT or TES1m LMP1 expression by 250ng/mL Dox for 24 hours. **(C)** Analysis of TRAF1, 2 and 5 roles in LMP1 association with cIAP1/2. Immunoblot analyses of 1% input vs anti-HA-LMP1 complexes immunopurified from Cas9+Daudi cells that expressed control versus TRAF1,2,3 or 5 targeting sgRNA and induced for WT or TES1m LMP1 expression by 250ng/mL Dox for 24 hours. **(D)** Analysis of LCL TRAF roles in LMP1 association with cIAP1/2. Immunoblot analyses of 1% input vs anti-FLAG-LMP1 complexes immunopurified from Cas9+LCLs that express FLAG-tagged LMP1 at physiological levels and that also expressed control versus TRAF1,2,3 or 5 targeting sgRNA. Anti-mouse IgG was used as a negative control. Blots are representative of n=3 (A and B) or 2 (C and D) experiments.

and by immunoblot observed robust cIAP1 polyubiquitination in lysates from LMP1 co-transfected control, TRAF2 KO or TRAF5 KO cells. By contrast, TRAF3 KO nearly completely ablated cIAP1 polyubiquitin chain decoration in LMP1-expressing cells (Fig 6A). Similar results were observed in TRAF3-edited Daudi Burkitt cells and LCLs (Fig 6B–6C). We also examined whether LMP1 induced ubiquitination of TRAF3. Interestingly, although LMP1 TES1 but not TES2 signaling promoted TRAF3 poly-ubiquitination, CD40-ligand stimulation triggered substantially higher levels of TRAF3 poly-ubiquitination (S5A Fig). Taken together, these findings support a model in which LMP1 TES1 recruits TRAF3 to trigger cIAP1 ubiquitination and cIAP1/2 proteasomal degradation (Fig 6D).

### cIAP1 or cIAP2 over-expression impair LMP1 mediated non-canonical NF-κB activity

Non-canonical NF-κB pathways culminate in processing of the p100 NF-κB precursor into the active p52 transcription factor subunit. The observation that LMP1 induced degradation of cIAP1/2 led us to hypothesize that cIAP1 depletion is essential for LMP1-mediated activation of non-canonical NF-κB signaling, akin to how SMAC mimetics activate non-canonical NF-κB [84] (S5B Fig). To test the model that cIAP1 loss is important for TES1 mediated non-canonical activity, we stably expressed control GFP versus cIAP1 in Daudi cells, and then mock induced or induced LMP1 expression for 24 hours. Immunoblot analysis revealed that enforced cIAP1 expression significantly diminished LMP1-driven NIK accumulation and p100:p52 processing (Fig 7A–7B). Likewise, LMP1-induced p100:p52 processing was significantly impaired by co-expression of either cIAP1 or cIAP2 in 293T cells (Fig 7C–7D), consistent with a model in which TES1 driven turnover of both cIAP1 and cIAP2 serves as a major driver of downstream non-canonical NF-κB activity. To further test this model, we performed time-course analysis. By 8 hours-post induction of wildtype but not TES1m LMP1, cIAP1 and cIAP2 were depleted, whereas NIK levels were increased (S5C Fig). We observed progressively increased p100:p52 processing indicative of non-canonical NF-κB activity beginning at 8 hours and which peaked at 12 hours of WT LMP1 expression (S5C Fig). Collectively, our data is consistent with a model in which LMP1 targets TRAF2, cIAP1 and cIAP2 for proteasomal degradation, which results in activation of downstream non-canonical NF-κB activity and release of active p52 transcription factor subunits for regulation of nuclear target gene expression (Fig 8).

### Discussion

Immune receptors and viral proteins utilize distinct mechanisms to initiate non-canonical NF-κB signaling, which has obligatory roles in B-cell differentiation and survival. Here, we present multiple lines of evidence that LMP1 TES1 signaling, critical for EBV-mediated primary B cell immortalization and for lymphoblastoid B-cell survival [23–27,60], triggers rapid depletion of the E3 ubiquitin ligases cIAP1, cIAP2 and TRAF2. This phenomenon was dependent on TRAF3, which binds tightly to the TES1 PxQxT motif [23–27,48,51,75], and which we found to recruit cIAP1 and cIAP2 to LMP1. LMP1 and TRAF3 together activated cIAP1 RING dependent ubiquitination and degradation of cIAP1 and cIAP2, as well as of TRAF2. cIAP1 depletion drove downstream NIK accumulation and p100:p52 processing, whereas enforced cIAP1 expression impaired LMP1-driven non-canonical activation.

It has been proposed that LMP1 activates non-canonical NF-κB signaling by sequestering TRAF3, thereby interfering with TRAF3 roles in basal NIK turnover [51]. However, only ~50% of TRAF3 is associated with LMP1 in LCLs [46,48,49],

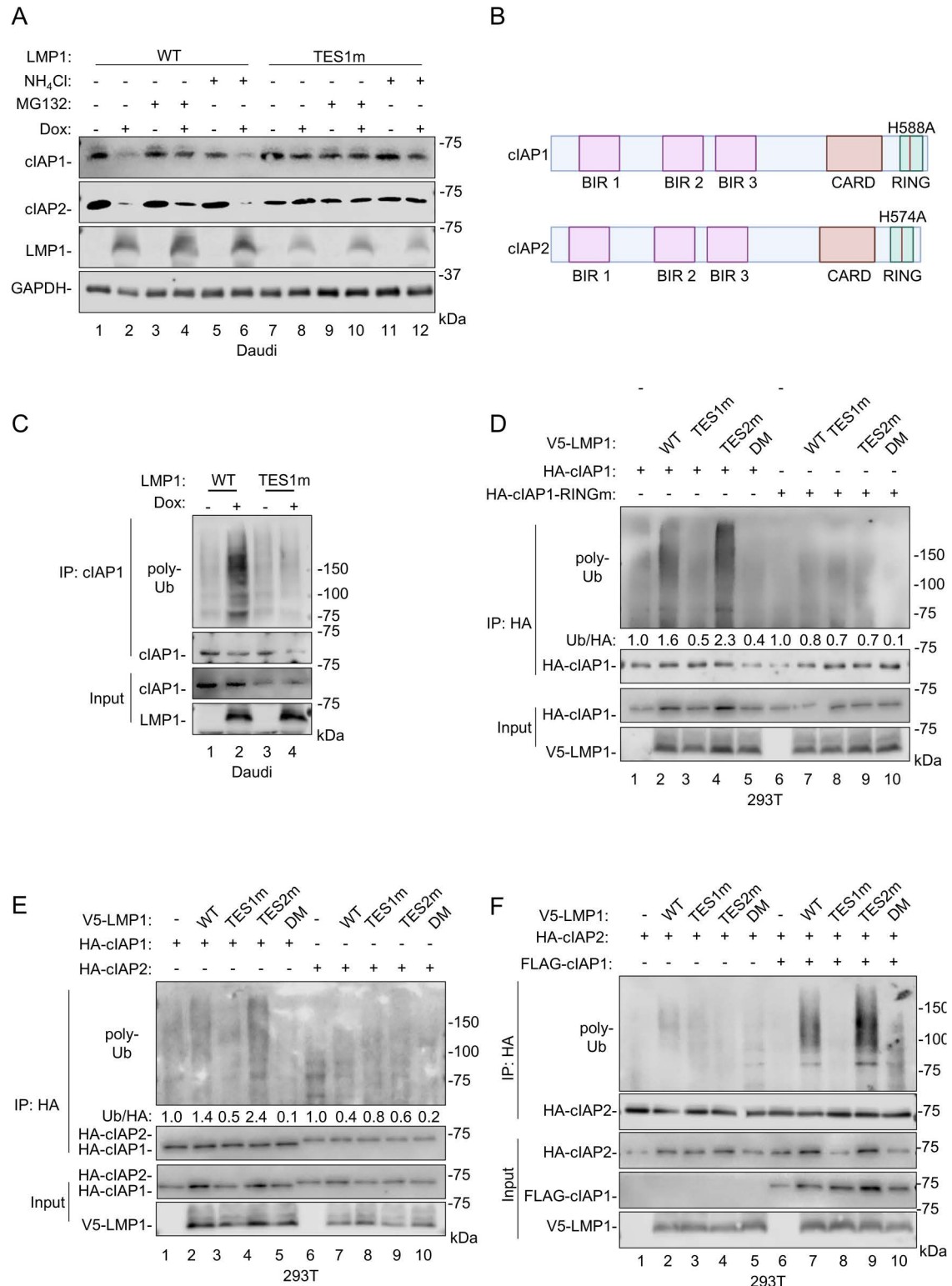

**Fig 5. LMP1 triggers cIAP1/2 polyubiquitination and proteasomal degradation. (A)** Analysis of proteasomal versus lysosomal roles in LMP1-driven cIAP turnover. Immunoblot analysis of WCL from Cas9+Daudi cells induced for WT or TES1m LMP1 expression by 250 ng/mL Dox for 24 hours, followed by treatment with the proteasome inhibitor MG132 (5μM) or the lysosomal acidification inhibitor NH$_4$Cl (20mM) for the last 8 hours. **(B)**

Schematic model of cIAP1 and cIAP2 domains. BIR, CARD and RING domains are highlighted. Created in BioRender. Sun, Y. (2026) https://BioRender.com/f2po9u4. **(C)** LMP1 TES1 signaling triggers cIAP1 polyubiquitination. Immunoblot analysis of 1% input vs endogenous cIAP1 immunopurified from Cas9 + Daudi cells induced for WT or TES1m LMP1 expression by 250 ng/mL Dox for 24 h. Cells were then treated with MG132 (5μM) for an additional 8 hours before collection. Cell lysates were boiled and immunoprecipitated with anti-cIAP1 antibody and protein A/G magnetic beads. **(D)** The cIAP1 RING domain is important for LMP1 TES1-driven cIAP1 polyubiquitination. Immunoblot analysis of input versus anti-HA-cIAP1, immunopurified from 293T cells that were transfected with expression vectors encoding HA-tagged cIAP1, cIAP1-RING mutant (RINGm, H588A) and empty vector or V5-tagged LMP1, as indicated. 24 hours following transfection, cells were treated with MG132 (5μM) for 24 hours prior to harvest. Cell lysates were boiled and immunoprecipitated with anti-HA-conjugated magnetic beads. Densitometry ratios of poly-ubiquitin versus HA-cIAP1 values are shown. **(E)** Analysis of LMP1-triggered cIAP1 vs cIAP2 polyubiquitination. Immunoblot analysis of input versus anti-HA-cIAP1 or anti-HA-cIAP2, immunopurified from 293T cells that were transfected with expression vectors encoding HA-tagged cIAP1 or cIAP2 and empty vector, V5-tagged LMP1, TES1m, TES2m or DM LMP1, as indicated. 24 hours following transfection, cells were treated with MG132 (5μM) for 24 hours prior to harvest. Cell lysates were boiled and then immunoprecipitated with anti-HA-conjugated magnetic beads. Ratios of polyubiquitin to HA-cIAP1 or cIAP2 IP are shown. **(F)** cIAP1 role in LMP1 TES1-driven cIAP2 poly-ubiquitination. Immunoblot analysis of input versus anti-HA-cIAP2, immunopurified from 293T cells that were transfected with expression vectors encoding empty vector, HA-tagged cIAP2, FLAG-tagged cIAP1, V5-tagged LMP1, TES1m, TES2m or DM LMP1, as indicated. 24 hours following transfection, cells were treated with MG132 (5μM) for 24 hours prior to harvest. Cell lysates were boiled and then immunoprecipitated with anti-HA-conjugated magnetic beads. Ratios of polyubiquitin to HA-cIAP2 IP are shown. Blots are representative of n = 3 (A,D,E) or 2 (C and F) experiments. To achieve comparable cIAP1 protein levels given LMP1 boosting of the expression vector promoter activity, three times the amount of cIAP1 vector was transfected into cells lacking LMP1 expression or expressing the LMP1 TES1m or DM mutants **(D-F)**.

raising the question of whether sequestration is sufficient for non-canonical NF-κB activation. Our findings complement this prior study by identifying that LMP1 not only tightly binds to TRAF3, but then uses TRAF3 to target cIAP1, cIAP2 and TRAF2 for degradation.

TRAF3 was originally identified as a NIK-binding protein in a yeast two-hybrid screen [88]. In the absence of stimuli, TRAF3 serves as an adaptor protein that enables cIAP1/2 to target NIK for ubiquitination and proteasomal degradation to prevent non-canonical NF-κB activation [41]. NIK ubiquitination involves a multi-subunit ubiquitin ligase complex composed of TRAF2, TRAF3, cIAP1 and/or cIAP2. Within this complex, TRAF2, but not TRAF3, directly interacts with cIAP1/2 [71]. Since cIAPs, as part of the TRAF–cIAP complex, are responsible for ubiquitinating and degrading NIK under resting conditions, their elimination disrupts this suppressive complex to activate non-canonical NF-κB signaling. Thus, in addition to TRAF3 sequestration, degradation of cIAP1/2 represents a critical step that amplifies LMP1-mediated activation of the noncanonical NF-κB pathway. This contrasts with the mechanism by which CD40 and BAFF receptors initiate non-canonical activity by targeting TRAFs 2 and 3 for proteasomal degradation to free NIK from cIAP1/2 targeting [65,66].

TRAFs 1, 2, 3 and 5 each associate with LMP1 TES1 within TRAF homo- or heterotrimeric complexes. Of these, TRAF3 binds most tightly to LMP1 [46,48,49,51]. In LCLs, only ~5% of the TRAF2 pool is associated with LMP1, in contrast with the ~50% of TRAF3 that is bound by LMP1 [46,48,49]. Furthermore, even though TRAF2 binds directly to cIAP1/2 [71,89], our data suggest that TRAF2 alone is not sufficient to recruit cIAPs to LMP1, since LMP1 targeted both cIAP1 and cIAP2 for degradation in TRAF2 knockout cells. It remains plausible that TRAF3-bound LMP1 serves as a scaffold for TRAF2/cIAP1/cIAP2 complexes, which are then routed to the proteasome. In support, we found that TRAF3 KO abrogated cIAP1/2 binding to LMP1. Alternatively, it is possible that functional redundancy between TRAF2 and other TRAFs, in particular TRAF5, may enable cIAP1/2 recruitment to LMP1 in a manner that is not absolutely dependent on TRAF2. Rigorous determination of the stoichiometry of TRAFs and cIAPs within the LMP1 complex will be an important goal for future studies.

A future goal will therefore be to identify how LMP1-bound TRAF3 activates cIAP1 ubiquitin ligase activity, autoubiquitination and proteasomal degradation. Small molecules that promote cIAP degradation called SMAC mimetics activate the non-canonical NF-κB pathway [90]. In some respects, TES1 signaling is reminiscent of SMAC mimetics, whose binding to cIAPs triggers rapid autoubiquitination and depletion. As observed with SMAC mimetics, cIAP depletion is dependent on cIAP1 RING ubiquitin ligase activity [91]. However, in contrast to SMAC, which similarly activates both cIAP1 and 2, our data suggest that cIAP1 ligase activity is critical and targets both cIAP1 and cIAP2 for degradation. Understanding how LMP1 TES1 signaling activates cIAP1 E3 ligase activity should reveal key details of cIAP regulation and may highlight potential therapeutic targets to block TES1 signaling. For instance, further molecular insights may allow the development

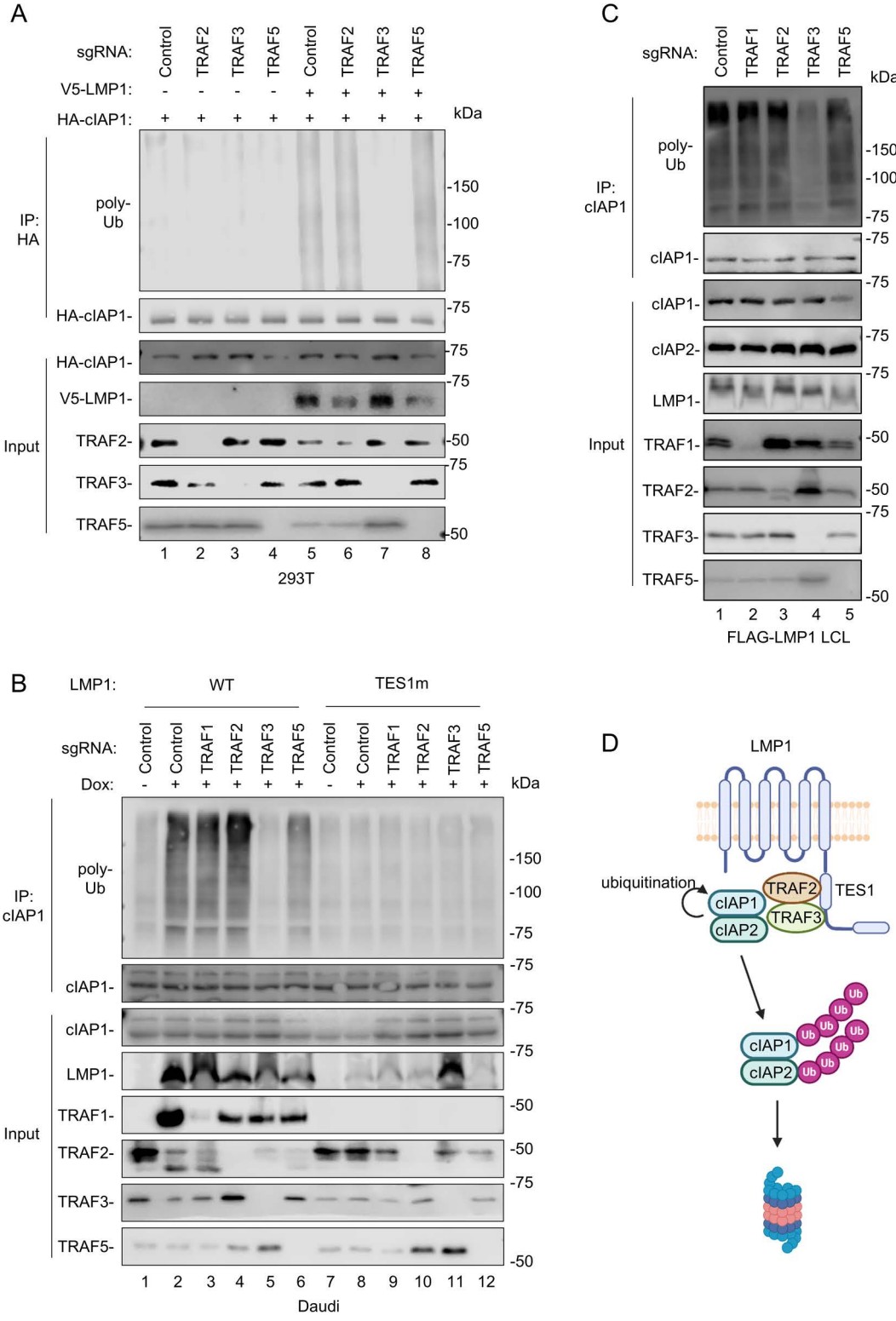

**Fig 6. TRAF3 is required for LMP1-induced cIAP1 ubiquitination. (A)** Analysis of TRAF roles in LMP-driven cIAP1 poly-ubiquitination in 293T cells. Immunoblot analysis of 1% input versus anti-HA-cIAP1, immunopurified from 293T cells that expressed control or TRAF targeting sgRNA and that were transfected with expression vectors encoding HA-cIAP1 and V5-LMP1 for 24 hours, as indicated. Cells were treated with MG132 (5 µM) for 24 h prior to harvest. Three times the amount of cIAP1 vector was transfected into cells when not co-transfected with LMP1 to achieve comparable cIAP1

protein levels, to compensate for LMP1 boosting of the expression vector promoter activity. Cell lysates were boiled prior to immunoprecipitation with anti-HA magnetic beads. **(B)** Analysis of TRAF roles in LMP-driven cIAP1 poly-ubiquitination in Daudi B-cells. Immunoblot analysis of input versus cIAP1 immunopurified from Cas9+Daudi cells that expressed control versus TRAF targeting sgRNAs and induced for LMP1 expression by 250ng/mL Dox for 24 hours. Cells were treated with 5 μM MG132 for 24 hours prior to harvest. Lysates were boiled prior to immunoprecipitation with anti-cIAP1 antibody and protein A/G magnetic beads. **(C)** Analysis of TRAF roles in LMP-driven cIAP1 poly-ubiquitination in LCLs. Immunoblot analysis of input versus cIAP1 immunopurified from Cas9+Daudi cells that expressed control versus TRAF targeting sgRNAs. Cells were treated with 5μM of MG132 for 24 hours prior to harvest. Cell lysates were boiled and immunoprecipitated with anti-cIAP1 antibody and protein A/G magnetic beads. Anti-mouse IgG was used as a negative control. **(D)** Schematic model of LMP1 TES1-induced cIAP degradation. LMP1 TES1 promotes autoubiquitination of cIAP1, which in turn ubiquitinates cIAP2 to trigger their rapid proteasomal degradation. Created in BioRender. Sun, Y. (2026) https://BioRender.com/7jkxwq5. Blots are representative of n=2 experiments.

of small molecules that perturb LMP1/TRAF3-mediated cIAP1/2 degradation. These would have the potential to selectively block LMP1, without inhibiting immunoreceptor signaling, including by CD40.

cIAP1 and cIAP2 are themselves NF-κB pathway targets and their mRNA abundances are upregulated by LMP1 [27,92,93]. In latency III B cells such as LCLs, LMP1-driven NF-κB activity drives cIAP1 and 2 re-syntheses. Thus, cIAP1/2 are continually expressed and degraded in LMP1+cells. Consequently, they can be detected at steady state levels by immunoblot of cells with stable LMP1 expression. However, CHX chase analysis revealed shorter $T_{1/2}$ in the presence of TES1 signaling. This may serve to sufficiently deplete local pools of cIAPs in order to remove their brake on the non-canonical NF-κB pathway, while leaving some residual pools to perform other essential cellular roles, such as in support of pro-survival signaling downstream of the TNF receptor [92,94,95]. In support of this point, LMP1 signals constitutively and independently of ligand. By contrast, immunoreceptors including CD40 typically signal for short bursts. LMP1 TES1 may therefore have evolved a unique mechanism to downmodulate, without completely suppressing, cIAP expression. We suspect that this allow LMP1+cells to initiate non-canonical NF-kB signaling, while maintaining steady state cIAP1/2 at levels needed to block apoptosis driven by death receptors, including TNF. However, it may be possible to devise strategies to further deplete cIAP1/2 in LMP1+cells in order to selectively sensitize EBV-infected cells to death receptor signaling.

It is difficult to prove that TRAF2 is dispensable for TES1 signaling, since its depletion induces non-canonical NF-κB activity. Interestingly however, we observed the appearance of a lower-molecular-weight band immunoreactive with anti-TRAF2 antibody upon conditional LMP1 expression, present also in LCLs and in MUTU III. This may represent a TRAF2 isoform or more likely a TRAF2 degradation product. We previously reported that LMP1 triggers K63-linked polyubiquitin chain attachment to TRAF2 [26], but this is typically associated with signal transduction rather than degradation. LMP1 may therefore induce partial TRAF2 cleavage in a manner not dependent on proteasome activity. Our data suggests that TES1 signaling triggers TRAF2 cleavage, likely within the N-terminal RING domain. Future objectives will be to identify the protease that induces TRAF2 cleavage, and to define whether its activity is necessary for the initiation of non-canonical NF-κB activity downstream of TES1.

In conclusion, we identified that LMP1 TES1 activates the non-canonical NF-κB pathway through targeted degradation of cIAP1/2 and TRAF2. This mechanism involves the assembly of a multi-protein complex mediated by LMP1 TES1 and TRAF3 and drives downstream p100 processing to the active p52 NF-κB transcription factor subunit. These findings suggest that it may be possible to develop small molecule inhibitors that selectively block the apparently unique mechanism by which LMP1 TES1 activates non-canonical NF-κB activity, which is critical for EBV-mediated B cell transformation into immortalized lymphoblastoid cells and for their survival.

## Materials and methods

### Ethics statement

Deidentified and discarded leukocytes, left over following voluntary platelet donation, were obtained from the Brigham and Women's Hospital Blood Bank following receipt of informed consent by the blood bank. Our non-human subjects research studies with these discarded cells were done under Mass General Brigham Hospital Institutional Review Board (IRB)

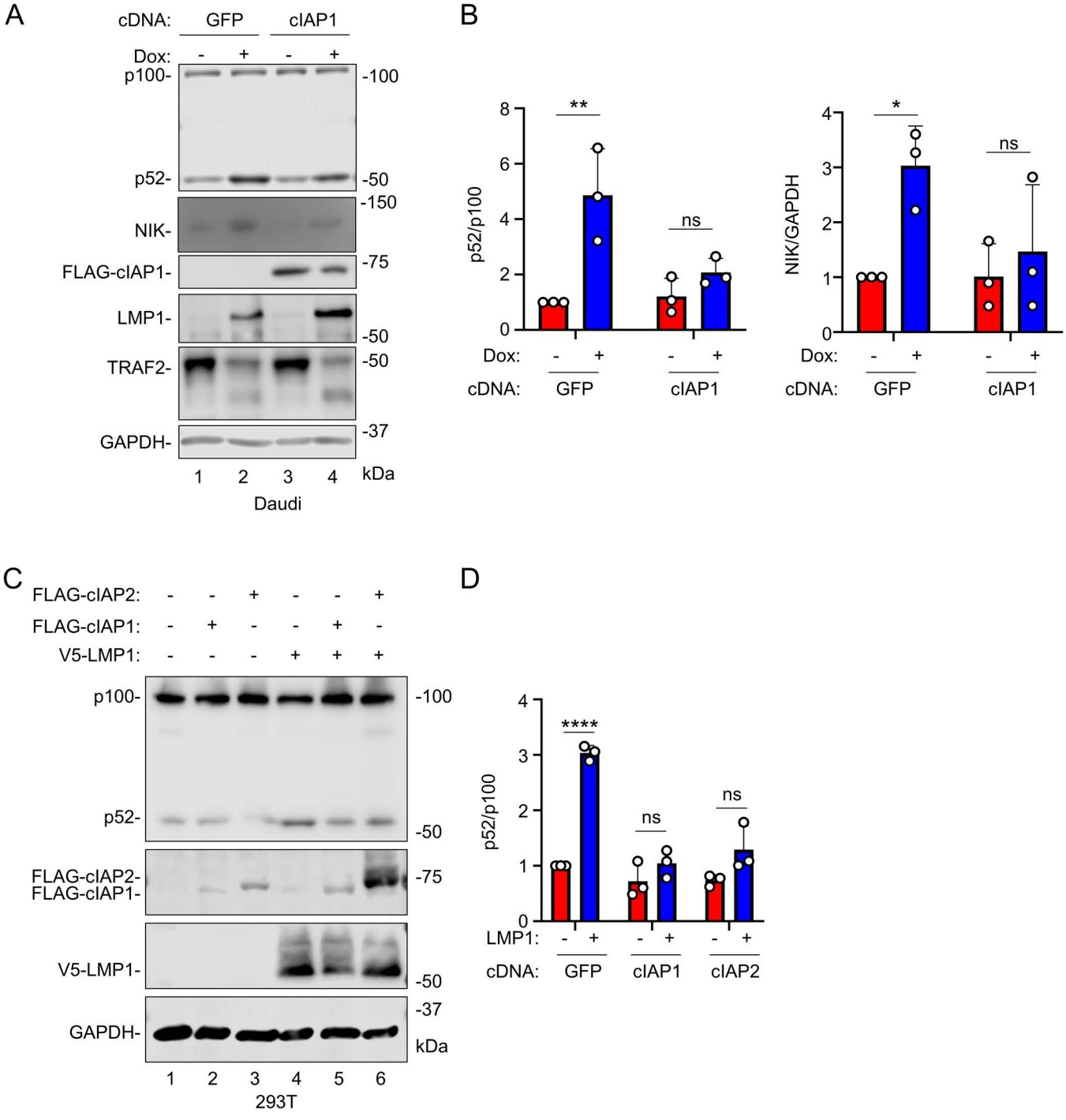

**Fig 7. cIAP1/2 overexpression impairs LMP1-induced non-canonical NF-κB pathway activation.** (A) Effect of cIAP1 over-expression on LMP1-induced p100 processing and NIK accumulation. Immunoblot analysis of WCL from Daudi cells transfected with control GFP versus cIAP1 expression vectors for 24 hours and then mock induced or induced for LMP1 expression by 250 ng/mL Dox for 24 hours, as indicated. (B) Quantification of p100:52 processing and NIK:GAPDH ratios from n=3 immunoblots, as in (A). Shown are the relative median p52:p100 ratios (left panel) and NIK:GAPDH ratios (right panel) + SD from n=3 replicates. Values in vehicle control treated GFP expressing cells were set to 1. (C) cIAP1 or cIAP2 overexpression effects on LMP1-induced non-canonical activity. Immunoblot analysis of WCL from 293T cells transfected with expression vectors encoding empty vector control, FLAG-cIAP1, FLAG-cIAP2 or V5-LMP1, as indicated for 24 hours. (D) Quantification of p52:100 ratios from n=3 immunoblots as in (C). Shown are the relative mean p52:p100 ratios + SD from n=3 replicates. Values in GFP expressing control cells were set to 1. Statistical significance was assessed by two-way ANOVA followed by Tukey's multiple comparisons test (B,D). ns, not significant, *p<0.05, **p<0.01. Blots are representative of n=3 experiments.

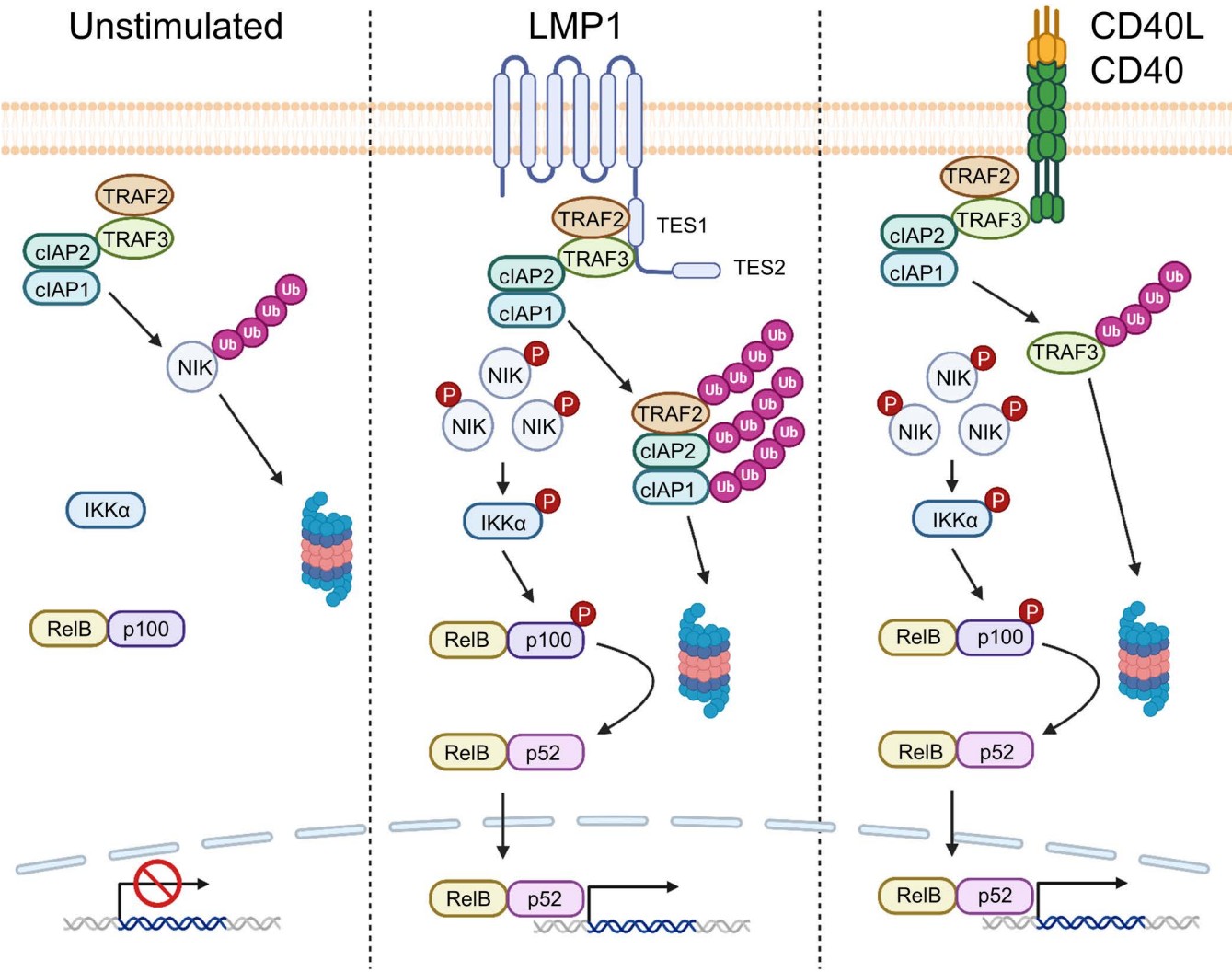

**Fig 8. Schematic model of LMP1 versus CD40 driven non-canonical pathways.** Under unstimulated conditions, the TRAF-cIAP complex targets the kinase NIK for proteasomal degradation, thereby preventing non-canonical NF-κB signaling. LMP1 TES1 triggers TRAF2, cIAP1 and cIAP2 for degradation, which initiates downstream signaling and p100:p52 processing. By comparison, CD40L/CD40 signaling induces degradation of TRAF3 to drive downstream non-canonical NF-kB pathway activation. Created in BioRender. Sun, Y. (2026) https://BioRender.com/wbt6q72.

approved protocols. Study approval number is: 2022P001270. Formal consent was obtained by the Brigham & Women's hospital blood bank before donation.

### Cell lines

HEK-293T were purchased from ATCC and cultured in Dulbecco's modified Eagle medium (DMEM, Gibco) supplemented with 10% fetal bovine serum (FBS, Gibco) and 1% penicillin/streptomycin (Gibco). Daudi and Akata Burkitt cells were obtained from Elliott Kieff. MUTU I and MUTU III Burkitt cells were obtained from Jeff Sample. FLAG-LMP1 LCLs were obtained from Elliott Kieff, generated using a recombinant EBV in which the FLAG tag encoding DNA sequence was knocked immediately upstream of the LMP1 ATG initiator codon [48]. All B cell lines were cultured in Roswell Park Memorial Institute (RPMI) 1640 (Life Technologies), supplemented with 10% v/v FBS and 1% pen/strep. B cell lines with stable

*Streptococcus pyogenes* Cas9 expression as previously described [96] were maintained with 5 µg/mL blasticidin (InvivoGen). Daudi and Akata cell lines with doxycycline-inducible WT, TES1m, TES2m, or DM LMP1 alleles were previously described [27]. Conditional LMP1 cells were maintained in either 50 µg/mL hygromycin or 3 µg/mL puromycin. For LMP1 induction studies, cells were seeded at $0.5 \times 10^6$ cells/mL and treated with 250 ng/mL doxycycline (Sigma) for 24 hours. All cells were incubated at 37°C with 5% $CO_2$.

### Primary human B cells

Peripheral blood mononuclear cells (PBMCs) were isolated using Lymphoprep Density Gradient Medium (Stem Cell Technologies), followed by negative selection of primary B cells using RosetteSep and EasySep Human B Cell Enrichment Kits (Stem Cell Technologies), in accordance with the manufacturers' instructions. Primary B-cells were grown in RPMI-1640 supplemented with 10% FBS.

### Construction of bacterial artificial chromosome (BAC) EBV LMP1 mutant genomes

The EBV p2089 BACmid was used for production of recombinant EBV genomes with wildtype, TES1 mutant or TES2 mutant genomes, as previously described [67]. P2089 encodes the EBV B95.8 strain genome, as well as cassettes encoding the prokaryotic F-factor, kanamycin resistance cassette, green fluorescence protein and Hygromycin B resistance cassette, as described previously [97]. BACmids were constructed using the GS1783 E. coli–based En Passant method previously described [98]. TES1 and TES2 point mutations were constructed using established BAC engineering protocols. For the TES1m EBV, 204 PQQAT was mutated to AQAAA to inactivate TES1 signaling. For the TES2m EBV, 384YYD was mutated to ID to inactivate TES2 signaling. BAC genome fidelity was validated by EcoRI, BamHI and NcoI DNA restriction map analysis, by DNA Sanger sequencing of high fidelity PCR-amplified TES regions, and by demonstration that revertants were fully transforming. BACmids were isolated and electroporated into spectinomycin-resistant BM2710 E. coli using 0.1 cm cuvette at 1.5kV, 200 Ohms, 25µF for infection of 293 cells. 293 producer cells were established by BM2710 *E. coli* invasion, as previously described [99]. Single cell 293 producer clones were screened.

### EBV stock preparation and quantification

EBV-positive 293 producer cells were reverse-transfected with expression plasmids encoding BZLF1 and BALF4 to induce the lytic cascade, as previously described [67]. Three days post transfection, supernatants were harvested and centrifuged at 1200 rpm for 10 mins and then at 4000 rpm for 10 mins to remove cell debris. Supernatants were passed through a 0.45µm filter to remove debris and concentrated by ultracentrifugation at 25000 rpm for 2 hours at 4°C. Viral stocks were tittered by the green Raji Assay [100]. In brief, $1 \times 10^5$ Raji cells were seeded in 100 µL per well of a 96-well plate and infected with serial dilutions of EBV. The percentage of GFP+ Raji cells was determined by flow cytometry 2 days post-infection. The concentration of infectious EBV virions was calculated as Green Raji Units per mL. For infection experiments, EBV was added at a multiplicity of infection (MOI) of 0.1 GRU per target cell.

### Antibodies and reagents

For immunoblot analysis, Cell Signaling Technology antibodies against cIAP1 (D5G9), cIAP2 (58C7), XIAP (3B6), TRAF1 (45D3), TRAF3 (#4729), TRAF5 (#D3E2R), FLAG (M2, #2368), V5 (#D3H8Q), NIK (#4994) and ubiquitin (P4D1, #3936) were used at 1:1000 for immunoblot analysis. Antibodies against TRAF2 (Proteintech 26846–1-AP), TRAF2 (Invitrogen SD205–06), HA (Abcam ab9110), and p100/52 (EMD Millipore #05–361) were used at 1:1000. Anti-GAPDH (EMD Millipore #MAB374) was used at 1:5000. The anti-LMP1 mouse monoclonal antibody S12 [101] was used from hybridoma supernatant at 1:1000. Cell Signaling Technology HRP-linked anti-mouse (#7076) and anti-rabbit (#7074) antibodies were used as secondary antibodies for immunoblot analysis at 1:5000. Proteintech anti-cIAP1 (1H3F1) and Abcam anti-HA (ab9110) were used for immunofluorescence assay at 1:200. Anti-FLAG M2 (Sigma) and Proteintech anti-cIAP1 (1H3F1)

were used for immunoprecipitation at 1:200. 100 µg/mL Cycloheximide (R&D systems), 5 µM MG132 (SelleckChem), 20 mM ammonium chloride (Sigma), 20 µM Birinapant (aka TL-32711) SMAC mimetic (SelleckChem) and 100 ng/ml CD40L (Enzo Life Sciences) were used for cell stimulation.

## CRISPR-Cas9 editing

CRISPR/Cas9 editing in B-cell lines with stable Cas9 expression was performed as previously described [60,96]. Briefly, Broad Institute pXPR-510 control sgRNA (targets a non-coding intergenic region), Avana, or Brunello sgRNA, as listed in Table 1, were cloned into pLentiGuide-puro (a gift from Feng Zhang, Addgene plasmid #52963), pLenti-spBsmBI-sgRNA-Hygro (a gift from Rene Maehr, Addgene plasmid #62205), or pLentiGuide-zeo (a gift from Rizwan Haq, Addgene plasmid #160091). The presence of sgRNA sequences were confirmed by Sanger sequencing. To generate lentivirus containing the sgRNA, 293T cells were transfected with pCMV-VSV-G (a gift from Bob Weinberg, Addgene plasmid #8454), psPAX2 (a gift from Didier Trono, Addgene plasmid #12260), and the sgRNA expression vector using the TransIT-LT1 transfection reagent (Mirus Bio). 293T supernatants were harvested and added to target B-cells at 48 and 72 hours post-293T transfection. As a control, target cells were transduced with sgRNA against GFP (pXPR-011, a gift from John Doench). Transduced cells were selected with puromycin (3 µg/ml), zeocin (200 µg/mL) or hygromycin (300 µg/mL). On-target CRISPR effects were validated by immunoblotting.

For CRISPR/Cas9 editing in HEK-293T cells, Brunello sgRNAs were cloned into pSpCas9(BB)-2A-Puro (PX459) V2.0 (a gift from Feng Zhang Addgene plasmid # 62988), which contains both Cas9 and cloning sites for sgRNA. Successful sgRNA ligation was confirmed by Sanger sequencing. 293T cells were transfected with the sgRNA-expressing PX459 vector using the TransIT-LT1 transfection reagent. Cells were not subjected to puromycin selection. Forty-eight hours post-transfection, cells were collected and on-target CRISPR effects were assessed by immunoblotting.

## SDS-PAGE and immunoblot

Immunoblot analyses were performed as described previously [102]. In brief, whole cell lysates were resolved by SDS-PAGE and transferred to nitrocellulose membranes (Bio-Rad), blocked with 5% nonfat dry milk in TBST for 1 h and incubated with primary antibodies at 4 °C overnight. Blots were washed three times in TBST and incubated with secondary antibodies for 1 h at room temperature. Blots were washed three times in TBST, incubated with ECL (Thermo Fisher #34578) and imaged by the Licor Fc platform. For quantification, densitometry data was obtained using the Image Studio Lite Ver 5.2 program.

## Immunoprecipitation

Immunoprecipitation was performed as previously described [103]. For B cells, 80 – 150 million cells were lysed in 1% v/v NP40, 150 mM Tris, 300 mM NaCl, 1 X EDTA-free protease inhibitor cocktail (Sigma), 1 mM $Na_3VO_4$ (Sigma) and 1 mM NaF (Sigma). 10% (v/v) of an aliquot from each lysate was preserved as the input. The rest of the lysate was incubated with a specific antibody targeting the epitope of interest overnight at 4°C with gentle rotation, followed by incubating with

**Table 1. sgRNA sequences used in this study.**

| sgRNA name | sgRNA sequence |
| --- | --- |
| Control | TTGACCTTTACCGTCCCGCG |
| TRAF1 | ACCTGGAGGTCGATTGCTAC |
| TRAF2 | ACTTGCCACAAGTCTTGACG |
| TRAF3 | CTTCTGCGAGAGCTGCATGG |
| TRAF5 | TGCATCCTGTCAGTTTCGAA |
| cIAP2 | TGGCTCTTATTCAAACTCTC |

Protein A/G magnetic beads for 2 hours. For anti-HA tag IP, 20 μL magnetic HA beads (Thermo Fisher) were added to the whole cell lysate. Following multiple washes with lysis buffer, proteins were eluted by boiling at 95 °C for 10 mins.

## Ubiquitination analysis

Cells were pretreated with MG132 for 24 hours before collection. 150 million B cells or 100 million HEK-293T cells were used. For HEK-293T cells, cells were transduced to express LMP1, cIAP1, and cIAP2 prior to MG132 treatment. To achieve similar levels of exogenous cIAP1/2 expression, one-third the amount of cIAP1/2 vectors was used in co-transductions with LMP1 compared to transductions performed without LMP1 or with LMP1 TES1m or DM. Site-directed mutagenesis using the NEB Q5 Site-Directed Mutagenesis Kit (E0554) was used to make the RING domain mutation in cIAP1. Cells were lysed with lysis buffer containing 1% v/v NP40, 150 mM Tris, 300 mM NaCl, 1 X EDTA-free protease inhibitor cocktail (Sigma), 1 mM $Na_3VO_4$, 1 mM PMSF (Thermo Fisher), 4 mM 1,10 o-phenanthroline (Sigma), 2 mM sodium pyrophosphate (Sigma) and 1 mM EDTA (Life Technology). 10% (v/v) of an aliquot from each lysate was preserved as the input. The remaining cell lysates were boiled at 95 °C for 5 mins, and then incubated with the specific antibody targeting the protein of interest overnight at 4°C with gentle rotation, followed by incubating with Protein A/G magnetic beads for 2 hours. Beads were washed with cold lysis buffer for four times, and then eluted by incubating at 95 °C for 10 mins. The eluted proteins were then subjected to immunoblot analysis.

## Inhibition of protein degradation

To inhibit proteasome-mediated protein degradation in LMP1 conditional expression cells, cells were treated with 5 μM MG132 for 8 hours following 24 hours of LMP1 induction. For inhibition of lysosome-mediated protein degradation, 20 mM ammonium chloride was added under the same conditions as MG132 treatment.

## Immunofluorescence analysis

Cells dried on glass slides were fixed in 4% paraformaldehyde (Santa Cruz) in PBS for 10 minutes, followed by permeabilization with 0.5% Triton X-100 (Sigma) in PBS for 5 minutes. Cells were blocked with 1% bovine serum albumin (Sigma) in PBS for 1 hour at room temperature. Cells were incubated with primary antibodies (1:200 dilution) in blocking buffer overnight at 4 °C. Following two washes with PBS, cells were stained with secondary antibodies. For detecting cIAP1 and LMP1, a cocktail of secondary antibodies conjugated with fluorophores were used with 1:1000 dilution in DPBS. For cIAP1, Goat anti-Mouse IgG Alexa Fluor 568 (H + L) secondary antibody (Invitrogen, # A-11031) was used, and for LMP1-HA, Donkey anti-Rabbit IgG Alexa Fluor 647 (H + L) secondary antibody (Invitrogen, # A-31573) was used. Cells were then washed twice with PBS and mounted overnight in ProLong Gold Antifade Mountant with DAPI (Thermo Fisher). Imaging and analysis were carried out using a Zeiss LSM 800 confocal microscope and Zeiss Zen Lite (Blue) software, respectively. To quantify the number of cells in which LMP1 and cIAP1 signals were colocalized, the following methods were used. The total number of cells in each field of view was first determined by single channel automated counting of the DAPI stained nuclei. Each field typically contained 10–20 cells. Colocalization analysis is performed by plotting the fluorescence intensity of each channel as a function of distance along the line. Colocalization of LMP1 and cIAP1 was determined by comparing the fluorescence intensity curves where overlapping of two peaks was judged as colocalization. Colocalization was then confirmed by ImageJ with the Colo2 plug-in. The % cells which have LMP1/cIAP1 colocalization was calculated by # cells which have colocalization/ # total cells in the field.

## Statistical analysis

All experiments were performed with two or three independent experiments. Statistical significance was assessed with Student's t test using GraphPad Prism 9 software, where NS = not significant, $p > 0.05$; * $p < 0.05$; ** $p < 0.01$; *** $p < 0.001$.

## Supporting information

**S1 Fig. LMP1 TES1 signaling downregulates cIAP1/2 and TRAF2 in Akata B cells.** (A) Analysis of LMP1 TES1 vs TES2 signaling effects on cIAP1/2 and TRAF levels. Immunoblot analysis of WCL from Akata cells induced for WT, TES1m, TES2m or DM LMP1 expression by 250 ng/mL Dox for 24 hours. Blots are representative of n = 3 experiments. (B) Relative fold changes + SD of GAPDH load-controlled cIAP1, cIAP2 or TRAF2 values, based on densitometry from n = 3 replicates of immunoblots as shown in (A). Values in vehicle control treated cells uninduced for WT LMP1 expression were set to 1. (C) Analysis of LMP1 and CD40L effects on cIAP1, cIAP2, and TRAFs expression in Daudi Burkitt cells. Immunoblot analysis of whole cell lysates (WCL) from Cas9 + Daudi Burkitt B-cells induced for WT or TES1m LMP1 expression by addition of 250ng/mL doxycycline (Dox) for 24 hours or treated with 50 ng/mL CD40L for 1 hour. p52:p100 ratios are indicated. Statistical significance was assessed by two-tailed unpaired Student's t-test (B). ns, not significant, *$p < 0.05$, **$p < 0.01$, ***$p < 0.001$.
(TIF)

**S2 Fig. TRAFs 1, 2 and 5 are dispensable for LMP1 induced-cIAP1/2 depletion.** (A) Immunoblot analysis of WCL from Cas9 + Daudi cells that expressed control or TRAF1 targeting sgRNA and that were induced for LMP1 WT or TES1m expression by 250ng/mL Dox for 24 hours. (B) Immunoblot analysis of WCL from Cas9 + Daudi cells that expressed control or TRAF2 targeting sgRNA and that were induced for LMP1 WT or TES1m expression by 250ng/mL Dox for 24 hours. (C) Immunoblot analysis of WCL from Cas9 + Daudi cells that expressed control or TRAF5 targeting sgRNA and that were induced for LMP1 WT or TES1m expression by 250ng/mL Dox for 24 hours. Blots are representative of n = 3 experiments.
(TIF)

**S3 Fig. LMP1 co-localizes with cIAP1 in a TES1 domain-dependent manner.** (A) Immunofluorescence analyses of cIAP1 and LMP1 localization in Daudi cells. WT, TES1m, TES2m or DM LMP1 expression was induced by 250ng/mL Dox for 24 hours, followed by treatment with 5 μM MG132 for 6 hours. Images are representative of 10 randomly chosen fields per sample. (B) Line scanning of cIAP1 (green) and LMP1 (red) fluorescence intensity within the annotated white rectangles shown in panel A. (C) Quantification of cells with overlapping cIAP1 and LMP1 signal in Daudi cells. Line scanning was performed with Zeiss Zen Lite (Blue) software. Statistical significance was assessed by one-way ANOVA followed by Tukey's multiple comparisons test. ns, not significant, ***$p < 0.001$, ****$p < 0.0001$.
(TIF)

**S4 Fig. cIAP proteasomal degradation is dependent on cIAP1 but not cIAP2.** (A) Relative fold changes + SD of GAPDH-normalized cIAP1 or cIAP2 levels based on densitometry from n = 3 replicates as in Fig 5A. Values in vehicle control treated cells uninduced for WT LMP1 were set to 1. Statistical significance was assessed by two-tailed unpaired Student's t test. ns, not significant, *$p < 0.05$, **$p < 0.01$. (B) Immunoblot analysis of WCL from Daudi cells induced for WT LMP1 expression by 250ng/mL Dox for 24 h, followed by treatment with 5μM MG132 for 8 hours. Blots are representative of n = 3 experiments. (C) Immunoblot analysis of WCL from Cas9 + Daudi Burkitt B-cells induced for WT, TES1m, TES2m, or DM LMP1 expression by addition of 250ng/mL Dox for 24 hours. TRAF2 blots were performed with antibodies raised against N-terminal or C-terminal residues. (D) Immunoblot analysis of WCL from Daudi cells induced for WT LMP1 expression by 250ng/mL Dox for 24 h or from HEK293T cells overexpressing truncated TRAF2 constructs. Created in BioRender. Sun, Y. (2026) https://BioRender.com/yp7zxhj. (E) Immunoblot analysis of input versus anti-FLAG-cIAP1, immunopurified from Daudi cells that were transfected with expression vectors encoding FLAG-tagged cIAP1, cIAP1-RING mutant (RINGm, H588A) and empty vector as indicated. Daudi cells were induced for WT or TES1m LMP1 expression by 250 ng/mL Dox for 24 h. Cells were then treated with MG132 (5μM) or with NH4Cl (20mM) for an additional 8 hours before collection. Cell lysates were boiled and immunoprecipitated with anti-FLAG antibody and protein

A/G magnetic beads. (F) Analysis of cIAP2 roles in LMP1 TES1-mediated effects on cIAP1, TRAFs and p52 expression. Immunoblot analysis of WCL from Cas9 + Daudi cells that expressed control vs cIAP2 targeting sgRNA and that were induced for LMP1 expression by Dox (250 ng/mL) for 24 hours. p52:p100 ratios are indicated.
(TIF)

**S5 Fig. LMP1 TES1 signaling downregulates cIAPs to activate non-canonical NF-κB.** (A) Immunoblot analysis of TES1 vs CD40 driven TRAF3 polyubiquitination. Daudi cells were transfected with an expression vector encoding FLAG-tagged TRAF3 and induced for WT or TES1m LMP1 expression by Dox (250 ng/mL) for 24 h or instead stimulated by Mega-CD40L (50 ng/mL) for 1 hour. Cells were then treated with MG132 (5µM) for an additional 8 hours. Cell lysates were boiled to disrupt complexes and immunoprecipitated with anti-FLAG antibody and protein A/G magnetic beads. (B) Cross-comparison of LMP1 versus SMAC mimetic effects on cIAP1, cIAP2, TRAF2 and TRAF3 levels in B cells. Immunoblot analysis of WCL from Cas9 + Daudi Burkitt B-cells induced for WT or TES1m LMP1 expression by addition of 250ng/mL Dox for 24 hours, or treated with the SMAC mimetic (SMACm) birinapant (20 µM) for 8 hours. p52:p100 and TRAF2:GAPDH ratios are indicated. (C) Kinetic analysis of LMP1 TES1 signaling effects on non-canonical NF-κB pathway activation. Shown are immunoblot analysis of WCL from Daudi cells induced for WT or TES1m LMP1 expression by addition of Dox (250ng/mL) for the indicated hours. GAPDH or p100 normalized densitometry ratios are shown below each lane, presented as relative levels observed in unstimulated cells.
(TIF)

**S1 Table. Combined raw data.**
(XLSX)

## Acknowledgments

We thank Makoto Ohashi and Eric Johannsen for valuable suggestions on the EBV BAC system. We thank Greg Smith for sharing the BM2710 bacteria and for valuable suggestions.

## Author contributions

**Conceptualization:** Yizhe Sun, Benjamin E Gewurz.

**Data curation:** Yizhe Sun, Shunji Li.

**Formal analysis:** Yizhe Sun, Shunji Li.

**Funding acquisition:** Yizhe Sun, Shunji Li, Bidisha Mitra, Benjamin E Gewurz.

**Investigation:** Yizhe Sun, Shunji Li, Bidisha Mitra, Ling Zhong, Aretina Zhang.

**Methodology:** Yizhe Sun, Shunji Li, Bidisha Mitra.

**Project administration:** Yizhe Sun, Benjamin E Gewurz.

**Resources:** Yizhe Sun, Shunji Li, Bidisha Mitra, Benjamin E Gewurz.

**Supervision:** Benjamin E Gewurz.

**Validation:** Yizhe Sun, Shunji Li.

**Visualization:** Yizhe Sun, Shunji Li.

**Writing – original draft:** Yizhe Sun, Shunji Li, Benjamin E Gewurz.

**Writing – review & editing:** Yizhe Sun, Shunji Li, Benjamin E Gewurz.

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
