## [Decision Letter · Decision Letter 0]

15 Oct 2025

PPATHOGENS-D-25-02231

Epstein-Barr Virus Latent Membrane Protein 1 targets cIAP1, cIAP2 and TRAF2 for Proteasomal Degradation to Activate the Non-canonical NF-kB Pathway

PLOS Pathogens

Dear Dr. Gewurz,

Thank you for submitting your manuscript to PLOS Pathogens. After careful consideration, we feel that it has merit but does not fully meet PLOS Pathogens's publication criteria as it currently stands. Therefore, we invite you to submit a revised version of the manuscript that addresses the points raised during the review process.

Please submit your revised manuscript within 30 days Dec 14 2025 11:59PM. If you will need more time than this to complete your revisions, please reply to this message or contact the journal office at plospathogens@plos.org. Please include the following items when submitting your revised manuscript:

We look forward to receiving your revised manuscript.

Kind regards,

Nancy Raab-Traub, Ph.D.

Academic Editor

PLOS Pathogens

Robert Kalejta

Section Editor

PLOS Pathogens

Sumita Bhaduri-McIntosh

Editor-in-Chief

PLOS Pathogens

orcid.org/0000-0003-2946-9497

Michael Malim

Editor-in-Chief

PLOS Pathogens

orcid.org/0000-0002-7699-2064

**Additional Editor Comments:**

The reviewers are in agreement that this is a significant study with convincing data. Each reviewer has specific suggestions and requests for additional experiments. Many of the concerns can be addressed with additional commentary and discussion. Concerns regarding loading controls and clarity of data should be addressed. Additionally, careful quantitation of components in co-precipitates would strengthen the conclusions. Considerations about the generality cell line to BL to other lymphocytes and to other EBV diseases can also be presented in the discussion or a modified introduction as suggested by Reviewer 3. The requested comparison to CD40 is interesting and you may already have some data analyzing this. The reviewers are very interested in the TRAF2 cleavage piece but recognize that it is not within the scope of the study.

**Journal Requirements:**

At this stage, the following Authors/Authors require contributions: Yizhe Sun, Shunji Li, Bidisha Mitra, Ling Zhong, Aretina Zhang, and Benjamin E Gewurz. Please ensure that the full contributions of each author are acknowledged in the "Add/Edit/Remove Authors" section of our submission form.

https://journals.plos.org/plospathogens/s/submission-guidelines#loc-parts-of-a-submission

5) We have noticed that you have uploaded Supporting Information files, but you have not included a list of legends. Please add a full list of legends for your Supporting Information files after the references list.

6) Please ensure that the funders and grant numbers match between the Financial Disclosure field and the Funding Information tab in your submission form. Note that the funders must be provided in the same order in both places as well.

**Reviewers' Comments:**

Reviewer's Responses to Questions

**Part I - Summary**

Reviewer #1: This is mechanistically detailed study on how EBV LMP1 TES1 activates the non-canonical NFκB pathway. The authors propose that LMP1, through TRAF3 recruitment, induces cIAP1 autoubiquitination and proteasomal degradation of cIAP1, cIAP2, and TRAF2, leading to p100→p52 processing. This mechanism is positioned as distinct from CD40/BAFF signaling. The study adds mechanistic detail to LMP1 signaling that has remained incompletely understood. The findings are solid, and conclusions are promising.

Reviewer #2: This manuscript investigates how the EBV-encoded latent membrane protein-1 activates the non-canonical NF-κB pathway. The authors provide compelling evidence that TES1 of LMP1 associates with TRAF3 to recruit cIAP1/2 and TRAF2, leading to their ubiquitination and proteasomal degradation. This process results in NIK stabilisation and downstream p100 to p52 processing. The work identifies a novel mechanism of viral manipulation of host signalling and highlights potential therapeutic targets.

The study is significant and timely. The experiments are carefully designed and well controlled. However, a few mechanistic gaps, translational validations, and presentation issues should be addressed before publication.

Reviewer #3: This manuscript by Sun, et al. addresses an important mechanistic question in EBV biology: how does LMP1 TES1 activate non-canonical NFκB signaling? The authors demonstrate that LMP1 TES1 recruits TRAF3, which serves as an essential adapter to bring cIAP1/2 ubiquitin ligases into proximity with LMP1, triggering their RING-dependent autoubiquitination and proteasomal degradation along with TRAF2. This degradation of negative regulators liberates NIK from constitutive degradation, enabling p100-to p52 processing and non-canonical NFκB activation.

The key finding that distinguishes this mechanism from CD40 and BAFF receptor signaling is that LMP1 preserves TRAF3 while targeting cIAP1/2 for degradation, whereas other immunoreceptors target TRAF3 itself. This resolves longstanding questions about how LMP1 activates non-canonical NFκB while requiring TRAF3, which was established as essential for LMP1 signaling by Xie et al in 2004. The work is well-controlled and uses appropriate genetic approaches including TRAF3 knockout B cells and cIAP1 RING domain mutants to demonstrate the mechanism. Overall, this represents a significant advance in understanding how this important viral oncogene usurps host signaling machinery.

**Part II – Major Issues: Key Experiments Required for Acceptance**

Reviewer #1: 1. The study rests on the premise that LMP1-mediated degradation of cIAP1/2 releases NIK from suppression. However, direct evidence of NIK stabilization, phosphorylation, or kinase activity is not shown in the setting.

2. The data suggest cIAP1 ligase activity is dominant, mediating degradation of both cIAP1 and cIAP2. Yet, the precise contribution of cIAP2 remains less clear. Selective depletion or complementation experiments could clarify whether cIAP2 is redundant, partially redundant, or required in specific contexts.

3. The appearance of a lower molecular weight TRAF2 species in Fig 1B, Fig2A, and S1A suggests partial cleavage in addition to degradation. The responsible protease(s) and its role in NFκB activation are not addressed.

Reviewer #2: 1) The manuscript reports a lower molecular weight TRAF2 species upon LMP1 expression, yet the mechanism and functional significance of this remains unclear. Does this represent proteolytic cleavage?

2) The majority of data are derived from Burkitt lymphoma cell lines, LCLs, and 293T overexpression assays. To enhance translational impact, the authors should, if feasible, test whether cIAP1/2 downregulation is detectable in EBV-positive tumour samples (e.g., Hodgkin lymphoma or NPC biopsies). Even limited immunohistochemical or immunoblot evidence would add significant weight.

3) Several ubiquitination experiments rely on HEK293T overexpression, which may not reflect physiological conditions. It would be helpful to confirm key findings (e.g., cIAP1 autoubiquitination) in EBV-positive B cells or LCLs to exclude cell-type specific artifacts.

4) The discussion emphasises similarity to SMAC mimetics but does not experimentally test this concept. Including proof-of-principle data using SMAC mimetics (or other cIAP antagonists) in LCLs would bridge the mechanistic insights with therapeutic relevance.

Reviewer #3: 1. Lack of mechanistic detail regarding TRAF3-mediated cIAP1 activation. The manuscript establishes that TRAF3 is essential for cIAP1 recruitment and subsequent autoubiquitination but doesn't fully explain how TRAF3 binding activates cIAP1's E3 ligase activity. What is the mechanism by which TRAF3 association with LMP1 triggers cIAP1 RING domain activation? Is this through proximity-induced dimerization, conformational changes, or relief of autoinhibition?

2. The paper should more explicitly address why TRAF3 remains stable while serving as an adapter for cIAP1/2 degradation, particularly given that cIAPs normally ubiquitinate TRAF3 in other signaling contexts. What protects TRAF3 from degradation in LMP1 complexes? The authors should analyze TRAF3 ubiquitination patterns in LMP1-expressing cells and consider whether LMP1 induces post-translational modifications that alter TRAF3's susceptibility to cIAP-mediated degradation. This mechanistic detail would strengthen the model and distinguish it more clearly from CD40/BAFF signaling.

3. The manuscript would benefit from more quantitative analysis of protein-protein interactions. The authors should determine the stoichiometry of TRAF3:cIAP1:cIAP2 within LMP1 complexes and provide more rigorous quantification of the co-immunoprecipitation experiments. Co-immunoprecipitation experiments should include quantitative controls and potentially mass spectrometry to define complex composition more precisely. Additionally, time-course experiments comparing the kinetics of cIAP1/2 degradation versus NIK accumulation would better establish the temporal sequence of events and strengthen the mechanistic model.

**Part III – Minor Issues: Editorial and Data Presentation Modifications**

Reviewer #1: 1. It would be appreciated if the authors can address this criticism: Study LMP1 signaling and EBV type III latency using EBV+ Burkitt lymphoma (BL) cell lines as models. However, EBV+ BLs in humans do NOT express type 3 latency since they have a stringent type I latency in which LMP1 is not expressed to any degree. The BL cell lines used by the investigators are due to a cell culture artifact ("type III latency") that occurs using certain EBV BL cell lines in tissue culture; BL cell lines with type III viral latency are not thought to be good models for actual human BLs. That said, the findings are not physiological relevant to BL. It is also unclear whether LMP1 shares the same mechanism in signaling transduction in BL and other EBV+ cancers.

2. Loading controls are missing in some blots where they are necessary.

3. Figures are generally clear, but some blots (e.g., TRAF2 cleavage bands) would benefit from higher resolution or inclusion of molecular weight markers.

4. The discussion could expand briefly on potential consequences of cIAP depletion for TNFR-mediated pro-survival signaling, as this might create vulnerabilities exploitable therapeutically.

5. The analogy to SMAC mimetics is intriguing, but no experiments directly test whether SMAC mimetics modulate LMP1 TES1 signaling in EBV+ B cells. Given the translational framing, inclusion of such experiments or a more cautious discussion would be appropriate.

Reviewer #2: Minor comments

1) The EBV latency section in the Introduction could be shortened to streamline focus on LMP1 and NF-κB signalling.

2) Figures 5 and 6 are dense. Consider adding schematic cartoons.

3) Figure legends could better highlight which results support TES1 vs TES2 dependency.

4) There is some redundancy between the Results and Discussion sections that could be reduced for conciseness (e.g., TES1 vs TES2 role is repeated multiple times).

5) Ensure consistent usage of “TES1/2” vs “CTAR1/2” throughout.

Reviewer #3: 1. Direct comparison with CD40 signaling in identical experimental systems would strengthen the claims about mechanistic distinctions. The authors could perform side-by-side experiments demonstrating how CD40 targets TRAF3 for degradation while LMP1 preserves TRAF3 but degrades cIAPs. This comparison would validate that the observed effects are truly LMP1-specific rather than only relevant in the experimental systems used. It would also be valuable to test whether other TRAF3-binding receptors can similarly recruit and activate cIAP1/2.

2. The low molecular weight TRAF2 bands are intriguing and could be further explored, though a bit beyond the scope of the manuscript. What is the specific protease responsible for this cleavage and/or does this processing affect other TRAF2 functions beyond non-canonical NFκB signaling? The authors could identify the cleavage sites through N-terminal sequencing or perform complementation studies with cleavage-resistant TRAF2 mutants. Understanding TRAF2 processing may reveal additional layers of LMP1's regulatory mechanism and could be important for interpreting the functional consequences of the pathway.

PLOS authors have the option to publish the peer review history of their article (what does this mean? ). If published, this will include your full peer review and any attached files.

**Do you want your identity to be public for this peer review?** For information about this choice, including consent withdrawal, please see our Privacy Policy .

Reviewer #1: **Yes:** Shunbin Ning

Reviewer #2: No

Reviewer #3: No

**Figure resubmission:**
---

## [Editor Report · Decision Letter 1]

13 Jan 2026

Dear Dr. Gewurz,

We are pleased to inform you that your manuscript 'Epstein-Barr Virus Latent Membrane Protein 1 targets cIAP1, cIAP2 and TRAF2 for Proteasomal Degradation to Activate the Non-canonical NF-kB Pathway' has been provisionally accepted for publication in PLOS Pathogens.

Best regards,

Nancy Raab-Traub, Ph.D.

Academic Editor

PLOS Pathogens

Robert Kalejta

Section Editor

PLOS Pathogens

Sumita Bhaduri-McIntosh

Editor-in-Chief

PLOS Pathogens

orcid.org/0000-0003-2946-9497

Michael Malim

Editor-in-Chief

PLOS Pathogens

orcid.org/0000-0002-7699-2064

The concerns of the reviewers have been clearly addressed with additional new high quality data that unequivocally support the proposed model and mechanism of action. In support of the model, the activity of LMP1 is now compared with CD40 signaling and SMAC mimetics. These experiments identify several distinct differences including LMP1 specific induced processing of TRAF2. This is also further analyzed to reveal the potential site of cleavage. The clear presentation and discussion of the exceptionally convincing data provide a new understanding of LMP1 signaling and activation of NFkB.
---

## [Editor Report · Acceptance letter]

Dear Dr. Gewurz,

We are delighted to inform you that your manuscript, "Epstein-Barr Virus Latent Membrane Protein 1 targets cIAP1, cIAP2 and TRAF2 for Proteasomal Degradation to Activate the Non-canonical NF-kB Pathway," has been formally accepted for publication in PLOS Pathogens.

Best regards,

Sumita Bhaduri-McIntosh

Editor-in-Chief

PLOS Pathogens

orcid.org/0000-0003-2946-9497

Michael Malim

Editor-in-Chief

PLOS Pathogens

orcid.org/0000-0002-7699-2064